# Gut microbiota promotes host resistance to low-temperature stress by stimulating its arginine and proline metabolism pathway in adult *Bactrocera dorsalis*

**Muhammad Fahim Raza, Yichen Wang, Zhaohui Cai, Shuai Bai, Zhichao Yao, Umar Anwar Awan, Zhenyu Zhang, Weiwei Zheng, Hongyu Zhang**  *

State Key Laboratory of Agricultural Microbiology, Key Laboratory of Horticultural Plant Biology (MOE), China-Australia Joint Research Centre for Horticultural and Urban Pests, Institute of Urban and Horticultural Entomology, College of Plant Science and Technology, Huazhong Agricultural University, Wuhan, People's Republic of China

* hongyu.zhang@mail.hzau.edu.cn

**Data Availability Statement:** The authors confirm that all data underlying the findings are fully available without restriction. All relevant data are

## Abstract

Gut symbiotic bacteria have a substantial impact on host physiology and ecology. However, the contribution of gut microbes to host fitness during long-term low-temperature stress is still unclear. This study examined the role of gut microbiota in host low-temperature stress resistance at molecular and biochemical levels in the oriental fruit fly *Bactrocera dorsalis*. The results showed that after the gut bacteria of flies were removed via antibiotic treatment, the median survival time was significantly decreased to approximately 68% of that in conventional flies following exposure to a temperature stress of 10˚C. Furthermore, we found that *Klebsiella michiganensis* BD177 is a key symbiotic bacterium, whose recolonization in antibiotic treated (ABX) flies significantly extended the median survival time to 160% of that in the ABX control, and restored their lifespan to the level of conventional flies. Notably, the relative levels of proline and arginine metabolites were significantly downregulated by 34- and 10-fold, respectively, in ABX flies compared with those in the hemolymph of conventional flies after exposure to a temperature stress of 10˚C whereas recolonization of ABX flies by *K. michiganensis* BD177 significantly upregulated the levels of proline and arginine by 13- and 10- fold, respectively, compared with those found in the hemolymph of ABX flies. qPCR analysis also confirmed that *K. michiganensis*-recolonized flies significantly stimulated the expression of transcripts from the arginine and proline metabolism pathway compared with the ABX controls, and RNAi mediated silencing of two key genes Pro-C and ASS significantly reduced the survival time of conventional flies, postexposure low-temperature stress. We show that microinjection of L-arginine and L-proline into ABX flies significantly increased their survival time following exposure to temperature stress of 10˚C. Transmission electron microscopy (TEM) analysis further revealed that low-temperature stress caused severe destruction in cristae structures and thus resulted in abnormal circular shapes of mitochondria in ABX flies gut, while the recolonization of live *K. michiganensis* helped the ABX flies to maintain mitochondrial functionality to a normal status, which is important for the arginine and proline induction. Our results suggest that gut microbiota plays a vital role

within the paper and its Supporting Information files. Genome sequence of Klebsiella michiganensis BD177 is available from NCBI BioProject (Accession number: PRJNA602959).

**Funding:** This research was supported by the National Natural Science Foundation of China (Nos. 31872931 and 31572008), the earmarked fund for the China Agricultural Research System (No. CARS-26), National Key R&D Program of China (No. 2017YFD0202000) and the Fundamental Research Funds for the Central Universities (No. 2662019PY055) to H.Z. The funders had no role in study design, data collection and analysis, decision to publish, or preparation of the manuscript.

**Competing interests:** The authors have declared that no competing interests exist.

in promoting the host resistance to low-temperature stress in *B. dorsalis* by stimulating its arginine and proline metabolism pathway.

## Author summary

The physiological mechanisms by which insects survive at low temperatures have not yet been discovered. In addition, there are no reports to verify the role of gut symbionts in the *B. dorsalis* biology during low-temperature stress. By combining transcriptomic and metabolomic approaches, we found that the presence of gut symbionts, especially *K. michiganensis* BD177, helps the host *B. dorsalis* to elevate the levels of particular cryoprotectant transcripts and metabolites, which suggests that the gut symbiont *K. michiganensis* BD177 stimulates the host arginine and proline metabolism pathway to promote its resistance to long-term low-temperature stress via influencing the mitochondrial functionality. Finally, the methods of this study are relevant for investigating the underpinnings of other host-microbe interactions during environmental stress.

## Introduction

Animals are inevitably challenged by adverse environmental conditions, such as extreme temperature, toxic substances or pathogen infections, ultraviolet (UV) radiation, insecticide exposure and oxidative stress [1, 2]. Environmental stress has a direct effect on metabolism and cellular processes that affect individual survival [3–5]. Among the abiotic factors affecting the physiology of terrestrial organisms, temperature is certainly the most critical. Extreme temperature can be detrimental to ectotherms or insect populations either through direct effects, which include temperature-induced cellular or tissue injury, functional restrictions or mortality and/or through indirect effects, e.g., activity constraints [6–9]. However, some invasive insect pests of fruit crops exhibit a unique form of phenotypic response [10, 11] to mild or even extreme temperature conditions [12, 13] or a rapid adaptation of thermal traits [14, 15] that may facilitate the survival of those introduced or alien species in novel environments. Unpredictable climatic conditions, sudden cold spells or milder winters are among the explanatory factors involved in the reduced mortality [16], distribution and dynamics of many insect pest species [17]. The mechanisms underlying cold acclimation responses are the main focus of much research in insect species [18, 19].

The gut microbiota provides nutrients to the host, such as amino acids, essential vitamins, carbon, and nitrogen compounds [20, 21]. Enterobacteriaceae species in the digestive tract of fruit flies are involved in nitrogen fixation [22], temporal host range expansion [23], reproductive success [21], detoxification [24], protection against pathogens and promoting host immunity [25]. The addition of the gut bacterial species *K. oxytoca* to the postirradiation diet resulted in improved sterile male competitiveness in Mediterranean fruit flies, and colonization of this strain in the host gut functions to suppress the proliferation of pathogenic strain *Pseudomonas* [26].

*Bactrocera dorsalis* (Diptera: Tephritidae) is one of the most invasive, polyphagous and multivoltine members of the Tephritidae family; this species causes substantial loss of cultivated crops throughout most of Asia and damages more than 250 host fruits and vegetables worldwide [27]. Survival under different temperature regimes is a critical factor to consider when *B. dorsalis* is accidently introduced into new areas, and this factor likely plays an

important role in the invasion process, i.e., overcoming a basic population establishment barrier. The population expansion of *B. dorsalis* indicates that it possesses high invasive potential due to its broad host range, relatively broad climate tolerance, and dispersal capacity [28, 29], posing considerable challenges to its effective management. Previous studies based on high-throughput pyrosequencing of the 16S rRNA gene and polymerase chain reaction-denaturing gradient gel electrophoresis (PCR-DGGE) fingerprinting have identified the prevalence of microbial communities occupying the gut and the reproductive organs of *B. dorsalis* [30, 31]. Enterobacteriaceae was identified as the predominant family, and many cultivable species were isolated from the gut of the host fly, such as *Enterobacter cloacae*, *K. oxytoca*, *Morganella sp.*, *Providencia rettgerii* and *C. freundii* [24, 31], indicating that they may play an important role in promoting host fitness under stressful conditions. The gut symbiont '*Citrobacter* sp.' of *B. dorsalis* has been reported to increase the host resistance against an organophosphate insecticide [24]. In our previous study, we determined that the intestinal probiotic *K. oxytoca* restored the ecological fitness of *B. dorsalis* following a decline induced by irradiation [32].

Herein, we examined the role of gut microbiota on host resistance to long-term low-temperature stress in insects. We found that removal of the gut microbiota significantly decreased the survival time of adult *B. dorsalis* following exposure to low-temperature stress, and reinfection of a key gut symbiont, *K. michiganensis* BD177, in the ABX flies restored the lifespan under low-temperature stress by stimulating the arginine and proline metabolism pathway in the host. These results suggest that the *B. dorsalis* gut microbiota helps host to increase temperature stress resistance via modulation of the arginine and proline metabolism pathway.

## Results

### Effect of the gut microbiota on the survival of *B. dorsalis* under low-temperature

ABX flies were generated via treatment with oral antibiotics. This treatment did not influence adult survival under normal rearing temperature ($27 \pm 1°C$) (S1A Fig) [33]. The efficacy of elimination of the gut microbiota was confirmed by plating gut homogenates onto Luria–Bertani (LB) agar plates (Fig 1A and 1B) and performing qPCR analysis (Fig 1C, $P = 0.0002$) using total gut bacteria gene primers (S1A Table). To investigate a possible role of gut microbiota in host resistance to temperature stress, ABX and conventional flies were subjected to different long-term low- to mild-temperatures stresses of $5°C$ (S1B Fig), $7.5°C$ (S1C Fig), $10°C$ (Fig 1D) and $15°C$ (S1D Fig); then, $10°C$ was chosen for the subsequent studies because this was the lowest temperature at which we found a significant difference in the median survival time between conventional and ABX flies (Fig 1D). In trial 1, the results showed that after removing the gut bacteria via antibiotic-treatment, the median survival time of ABX flies was significantly reduced to 6 days from 19 days in conventional flies ($P < 0.0001$, log rank test) following exposure to continuous low-temperature ($10°C$), which represents an approximately 68% decrease (Fig 1D), suggesting that an improvement in the host lifespan at $10°C$ may be linked with the intestinal microbial community.

### A key symbiotic bacterium, *K. michiganensis*, enhances the resistance of *B. dorsalis* to low-temperature stress

To find key symbiotic bacteria to improve the host survival under low-temperature stress, we used a culture-dependent approach and identified 223 and 217 isolates from conventional fly guts 5 days postexposure (5 dpe) to $28°C$ and $10°C$, respectively. In total, 15 representative bacterial species were identified. The most prevalent members of Enterobacteriaceae, such as

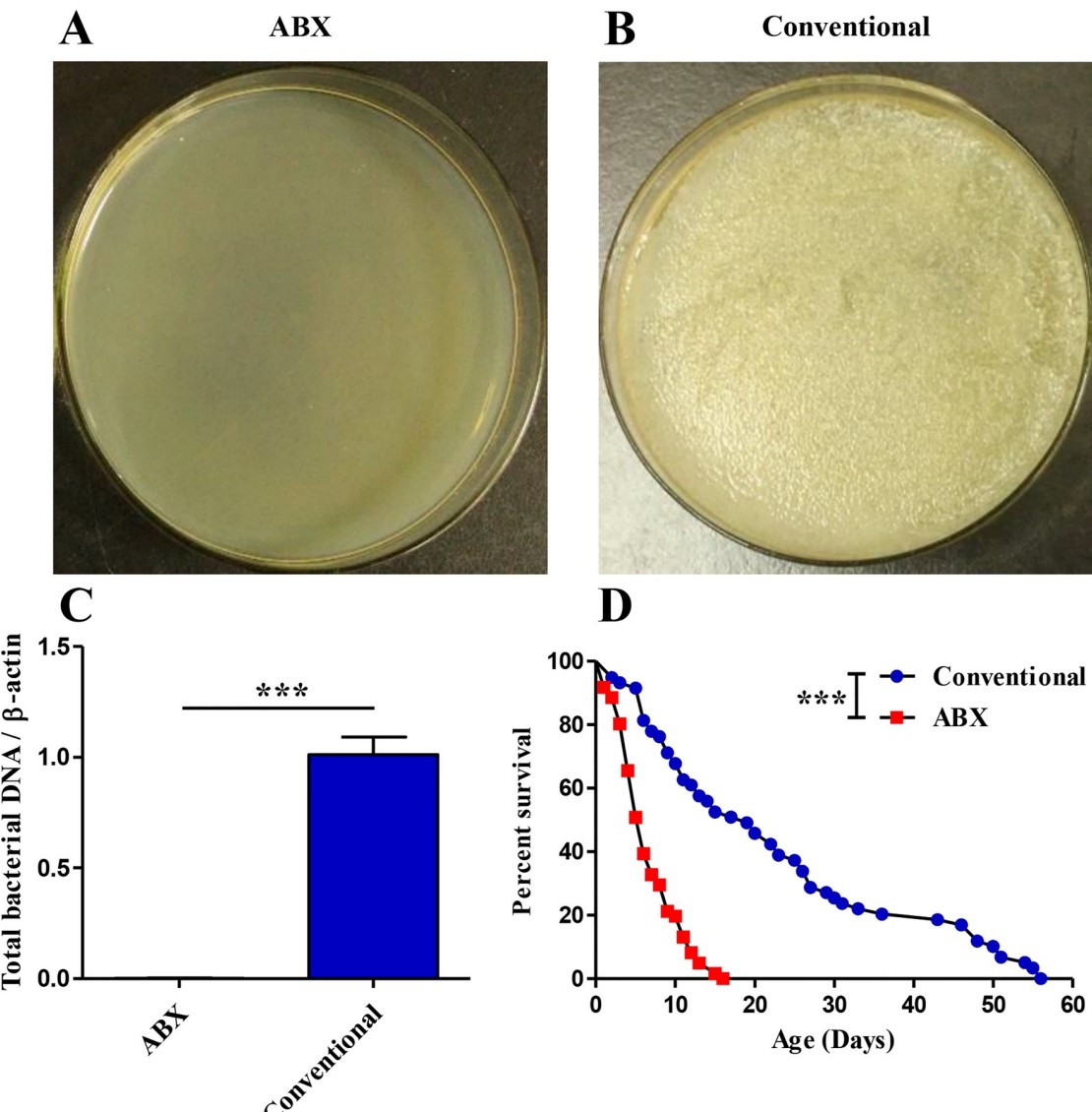

**Fig 1. Gut microbiota promotes the median survival time of *B. dorsalis* under low-temperature (Trial 1).** Generation of ABX flies (A- C). The efficacy of elimination of gut bacteria confirmed by culturing ABX (A) and conventional (B) fly (n = 10) gut homogenates on LB agar plates, and (C) performing qPCR analysis on the fly gut (n = 15) using universal gene primers. Three biological repeats were conducted. Error bars indicate mean with SE. Significant difference was determined by Student's t test at $P < 0.0001$. (D) Survival curves of conventional and ABX *B. dorsalis* by log-rank analysis postexposure to 10˚C (individual = 59–61 in each treatment); the median survival time of ABX flies was significantly reduced to 6 days from 19 days in conventional flies, *** $P < 0.0001$.

*Enterobacter* spp., *K. michiganensis*, *C. koseri*, *Kluyvera ascorbata*, and *Providencia* spp., as well as minor bacterial species such as *Acinetobacter* spp. were identified from *B. dorsalis* gut (S1B Table). The relative abundances of four major strains—*C. koseri*, *K. michiganensis*, *E. soli* and *E. hormaechei*—in conventional flies postexposure to 10˚C slightly decreased, although the differences were nonsignificant, by 28.57%, 12.12%, 10.53% and 9.09%, respectively, compared with those in conventional flies postexposure to 28˚C (S1B Table).

In trial 2, ABX flies were reinfected with all isolated gut bacterial species to determine whether the key gut bacteria strain could restore the ecological fitness parameter of host's

survival, postexposure to 10°C. The results showed that ABX flies reinfected with live *K. michiganensis* BD177 significantly extended the median survival time to 13 days from 5 days in ABX counterparts, which represents an approximately 160% increase ($P<0.0001$, Fig 2A). The median survival time was also significantly increased in ABX flies recolonized by major gut symbionts such as *E. soli* and *C. koseri* (11 days), *E. tabaci* (9 days), and *E. hormaechei* (8.5 days) compared with 5 days in ABX control but to a lower extent than that achieved with *K. michiganensis* reinfection (S2A Fig). Moreover, other minor gut strains extended the median survival time, including *A. radioresistens* and *P. alcalifaciens* (8 days), *Leclercia adecarboxylata* (7.5 days), *Enterococcus faecium* and *P. vermicola* (7 days), *P. rettgeri* (6.5 days) and *A. bereziniae* (6 days), beyond 5 days in the ABX control but to a significantly lesser extent than that achieved in *K. michiganensis*-reinfected flies (S2A Fig). In contrast, survival was unaffected after the recolonization of ABX flies by *Lactococcus garvieae*, *Serratia marcescens* and *K. ascorbata* compared with that in the ABX control (S2A Fig). Importantly, there was no significant difference in the median survival time between conventional (17.5 days) and *K. michiganensis*-reinfected (13 days) flies ($P = 0.1275$, Fig 2A), while all other tested strains exhibited a significantly decreased median survival time compared with that of conventional flies ($P<0.05$, S2A Fig), suggesting that *K. michiganensis* is a key gut symbiont that has greater effect on the restoration of the survival time of ABX flies, similar to the outcome achieved by conventional flies.

A single dose reinfection or recurring supply with heat-killed (HK) *K. michiganensis* equivalent to the levels of live *K. michiganensis* ($10^8$ CFU/ml) showed no significant effect on the survival time of ABX flies (Fig 2A, $P>0.05$) or in another independent longevity test (Fig 2B, $P>0.05$) compared with ABX control postexposure to 10°C. Moreover, increasing the quantity of HK *K. michiganensis* ($10^9$ CFU/ml) given to ABX flies throughout life did not even rescued the median lifespan (5.5 days) over the axenic control (5 days), and to a significantly lesser extent ($P < 0.0001$) than that of live *K. michiganensis* inoculation (Fig 2A). Taken together, our results show that survival of flies cannot be mimicked by HK *K. michiganensis*, suggesting that microbes may stimulate indirect processes or host metabolism, rather than directly providing nutritional benefits to the host during low-temperature stress.

Since a single reinfection of live *K. michiganensis* was sufficient to rescue the ABX host survival at 10°C, we next investigated whether, once inoculated, the host is able to maintain an adequate supply of gut microbes to benefit health throughout life postexposure to 10°C (S2B and S2C Fig). The results showed that after 10 dpe to 10°C, the average number of cultivated microbial communities resulting from colony forming units (CFUs) in ABX control flies fed a sterile liquid diet were 7.667 x $10^5$ ± 2.028 x$10^5$ CFUs gut$^{-1}$ (mean ± SE of 10 individual flies), which represents approximately 96% decrease versus that of live *K. michiganensis*-reinfected flies, which was 1.937 x $10^7$ ± 3.014 x $10^6$ CFUs gut$^{-1}$ ($P<0.005$, S2C Fig). Importantly, CFUs of flies after 10 dpe to 10°C; the average numbers of cultivated microbial communities resulting from CFUs in ABX flies fed with all other tested strains showed a significant decrease compared with live *K. michiganensis*-reinfected flies ($P<0.05$, S2C Fig), suggesting that, once inoculated, *K. michiganensis* can sustain in the ABX flies gut under low-temperature stress environment than that of all other tested strains. In contrast, there was no significant difference in the average number of cultivated microbial communities between conventional (1.98 x $10^7$ ± 2.38 x $10^6$ CFUs gut$^{-1}$) and live *K. michiganensis*-reinfected flies (1.937 x $10^7$ ± 3.014 x $10^6$ CFUs gut$^{-1}$) ($P>0.05$, S2C Fig). These results suggest that the survival of flies under low-temperature stress is influenced by the abundance of associated bacteria. Next, qPCR analysis also revealed that the microbial load of *K. michiganensis*, as well as total bacteria and Enterobacteriaceae in the gut homogenates, was maintained at two different time intervals during the fly survival trial (10 dpe and 20 dpe to 10°C) after feeding live *K. michiganensis*. (S2D–S2F Fig). Collectively, the microbial load in the gut of ABX flies reinfected with live *K.*

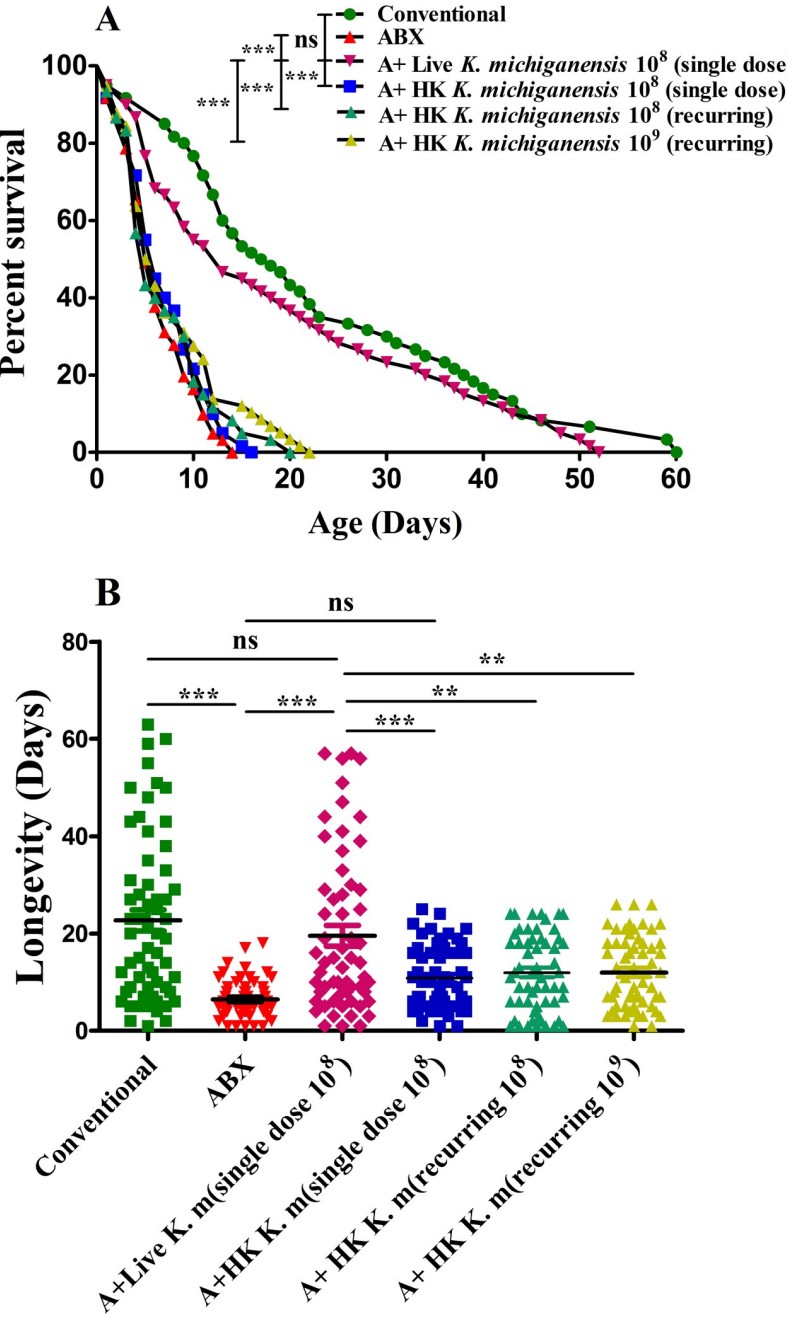

**Fig 2. Trial 2. Live *K. michiganensis* recolonization extends ABX flies survival time and longevity under low-temperature stress.** (A) Recolonization of ABX flies by live *K. michiganensis* ($10^8$ CFU/ml) extends survival (log-rank test, $P<0.0001$). ABX flies reinfected with HK *K. michiganensis* (provided a single dose and recurring quantity of $10^8$ CFU/ml) did not promotes survival (log-rank test, $P>0.05$) compared to the ABX flies under temperature stress of 10˚C. The median survival time was not significantly different between ABX flies reinfected with an increased HK *K. michiganensis* dose (5.5 days) (recurring supply of $10^9$ CFU/ml at every food change throughout life) and ABX flies (5 days), under temperature stress of 10˚C (n = 58–61 for each condition, log-rank test, $P>0.05$). (B) The scatter diagram of an independent trial of longevity of conventional flies, ABX flies fed with live *K. michiganensis*, single dose or recurring supply of HK *K. michiganensis* and the ABX control under temperature stress of 10˚C (n = 60 for each condition, one-way ANOVA followed by Tukey's multiple comparison test was performed with a significant difference at ***$P<0.0001$, ** $P<0.001$, ns $P>0.05$). *A+ strain name (ABX fly reinfection with above gut bacterial strains). In figure B, *K. m* indicates *K. michiganensis*.

*michiganensis* was similar to that of conventional flies (*P*>0.05) and significantly higher (*P*<0.005) than that of ABX control, suggesting that live *K. michiganensis* reinfection can form a sustained colony in the ABX flies gut that, in turn, improves host resistance to low-temperature stress.

Furthermore, we investigated the physiological mechanism, e.g., the major nutrients level in the hemolymph, in the presence or absence of gut microbes. The results showed that major nutrient levels, i.e., total amino acid (AA) (Fig 3A) and lipids (Fig 3B), were significantly decreased (*P*<0.05) by 69.67% and 85.43% in the hemolymph of ABX flies compared with those in conventional flies 5 dpe to 10˚C, respectively. Surprisingly, the sugar levels (Fig 3C) were not significantly different between these two treatments (*P*>0.05). After the feeding of live *K. michiganensis* to ABX flies, the total AA (Fig 3A, *P*<0.05) and lipid (Fig 3B, *P*<0.05) levels were restored and significantly enhanced by 170% and 300% in the hemolymph compared with those in their ABX counterparts, respectively. There were no significant differences in the major nutrient levels (AAs and lipids) between ABX flies reinfected with *K. michiganensis* and conventional flies (Fig 3).

**Influence of gut bacteria on the metabolomic response of the host to low-temperature stress.** For a more in-depth understanding of the mechanism whereby the gut microbiota promotes host resistance to low-temperature stress, we compared the metabolomics of the hemolymph for the conventional flies, ABX flies reinfected with *K. michiganensis* (both termed as 'nonABX flies') and ABX flies. Overall, several metabolites displayed considerable differences across the nonABX and ABX fly hemolymph, including amino acids, multiple lipids and their derivatives, and carbohydrates 5 dpe to 10˚C (S1A and S1B Data). In the principal component analysis (PCA) of these three hemolymph metabolite groups, PC1 explained 36.03% of the variance in expression and clustered conventional and *K. michiganensis*-reinfected flies separately from ABX flies, indicating that the gut microbiota has a remarkable effect on the hemolymph metabolomic profile at 10˚C (Fig 4A).

In the present study, histidine was the most significantly upregulated compound in *K. michiganensis*-reinfected flies (18-fold), while in conventional flies, it was upregulated by 31-fold compared to that of ABX flies (S1C and S1D Data), suggesting that this metabolite is absorbed by the host under low-temperature stress. The main constituents of arginine and proline metabolism were significantly enriched in the hemolymph of conventional or *K. michiganensis*-reinfected flies compared with those in ABX control (Fig 4B–4D). Notably,

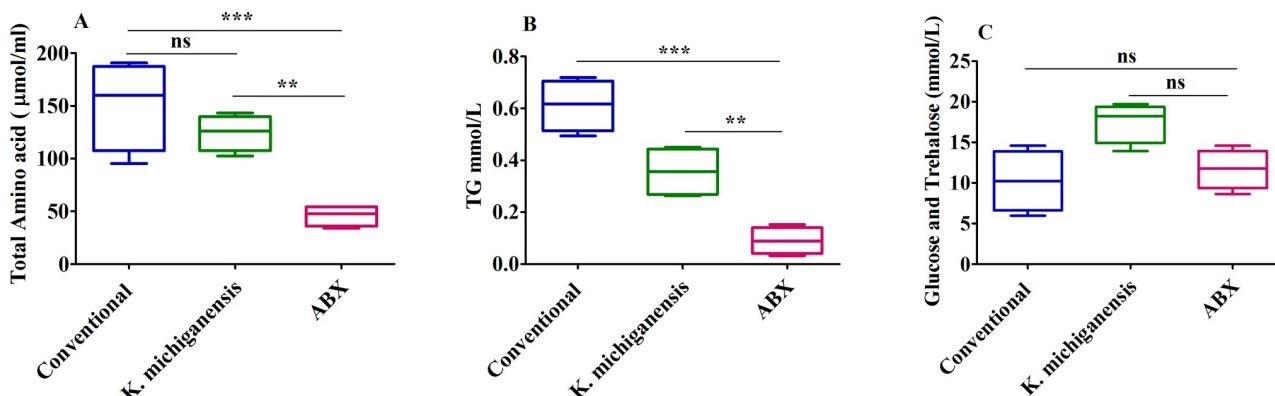

**Fig 3. Amino acid (A), lipid (B) and Sugar (C) levels after 5 dpe to 10˚C.** Ten flies (sex ratio 1:1) in each repeat (n), n = 4 were used. One-way analysis of variance (ANOVA) and Tukey's test were performed (*P*<0.05; and not significant denoted by ns, *P*>0.05).

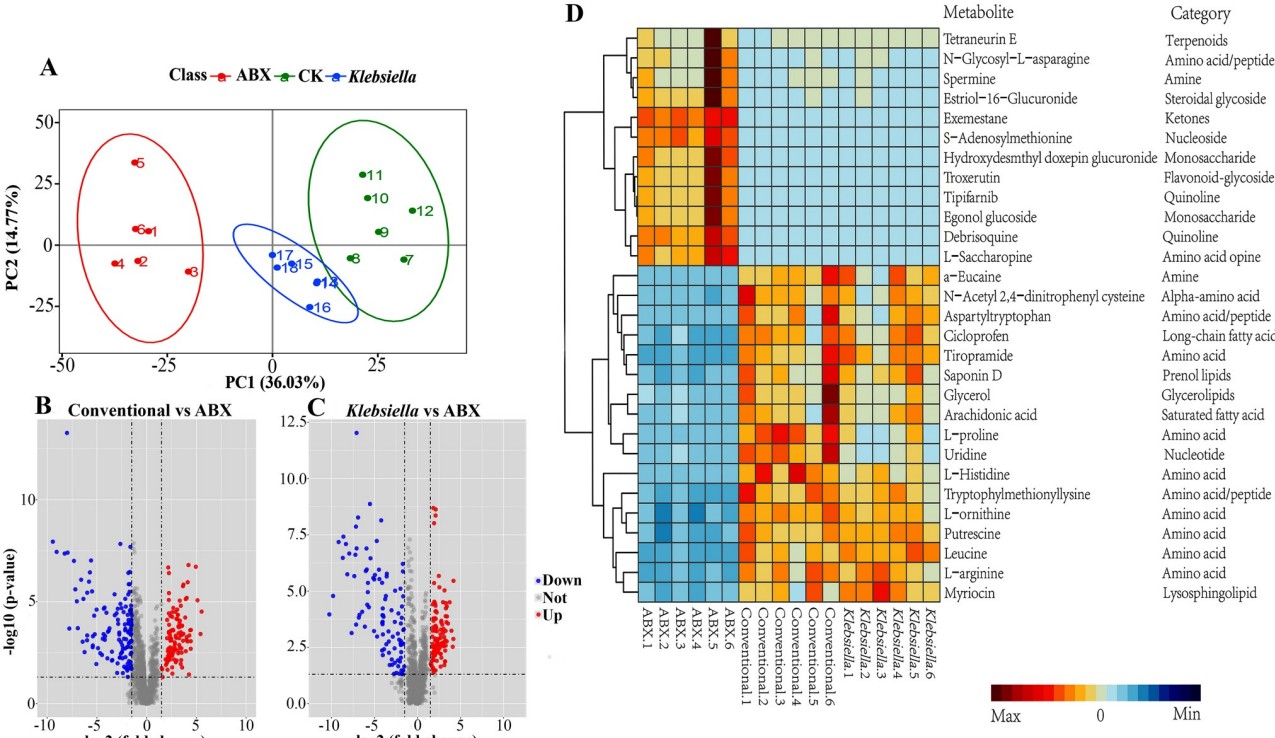

**Fig 4. Metabolomic analysis.** (A) Results of PCA based on 1548 metabolites detected from hemolymph of ABX (n = 1–6, red), CK (conventional) (n = 7–12, green) and *K. michiganensis*-reinfected (n = 13–18, blue) flies showing different clustering groups (95% confidence regions). (B) Volcano plots to infer the overall distribution of 201 differentially expressed metabolites in conventional vs. ABX fly hemolymph, (C) 202 differentially expressed metabolites in *K. michiganensis*-reinfected vs. ABX fly hemolymph. (D) Unsupervised hierarchical clustering heat map of the 25 metabolites that contributed most to the separation of hemolymph across different samples of ABX (n = 6), conventional (n = 6) and *K. michiganensis*-reinfected flies (n = 6). Each column represents one sample. Colors specify the relative concentration of each metabolite (minimum–maximum). The tree on the left illustrates a dendrogram of clustering (Ward's method).

metabolites, mainly including proline, arginine, ornithine and putrescine, were significantly downregulated by 34-, 10-, 5- and 4-fold in ABX flies compared with conventional flies, respectively, 5 dpe to 10˚C (S1C Data). Reinfection of ABX flies with *K. michiganensis* upregulated the relative levels of proline, arginine, ornithine and putrescine by 13-, 10-, 5- and 4-fold the levels found in ABX flies, respectively, 5 dpe to 10˚C (S1D Data). These findings suggest that cryoprotectants of arginine and proline metabolism elevated by the host in the presence of gut symbionts, in particular *K. michiganensis*, play a role in low-temperature stress resistance. Most of the differentially expressed metabolites (DEMs) were assigned to metabolic (amino acid, fatty acid synthesis) reference pathways, with arginine and proline metabolism being the top enriched pathway (S1E and S1F Data).

**Impact of gut bacteria on the transcriptomic response of the host to low-temperature stress.** To assess the effect of gut microbiota on host transcriptome at 10˚C, we employed RNA sequencing to identify the differentially expressed genes (DEGs). A total of 10,760 (7737 up- and 3023 downregulated) DEGs between *K. michiganensis*-reinfected vs. ABX samples and 18,576 (11,507 up- and 7069 downregulated genes) DEGs in conventional vs. ABX groups were identified. Several DEGs encoding heat shock proteins (*HSPs*), zinc finger proteins (*ZFPs*) and serine/threonine protein kinase (*STKs*) were enriched in the current study (S2A and S2B Data). To confirm the expression profiles identified from RNA-seq data, we performed qRT-PCR on several randomly selected DEGs. The comparative analysis revealed that

RT-qPCR-detected expression patterns supported the RNA-seq results, with consistent results in both comparisons, i.e., conventional vs. ABX (S3A Fig) and *K. michiganensis* vs. ABX groups (S3B Fig). Importantly, 36 of the 146 ($P<0.05$, 25% DEG:TEG) genes in *K. michiganensis*-reinfected vs. ABX flies and 56 of the 146 genes ($P<0.005$, 38% DEG:TEG) in conventional vs. ABX flies that mapped to the 'arginine and proline metabolism pathway' were significantly upregulated 5 dpe to 10˚C (S2C and S2D Data).

## Effect of gut microbiota on the regulation of the transcriptomic and metabolomic pathways of the host

To emphasize our efforts on a smaller subset of highly expressed genes and metabolites, we pooled significant overlapping pathways with those enriched in both the transcriptomic and metabolomic approaches as described by MacMillan et al. [34]. Among the top enriched pathways, only one pathway, 'arginine and proline metabolism,' in *K. michiganensis*-reinfected vs. ABX and conventional vs. ABX flies was shared across both 'omics' approaches. The observation that arginine and proline metabolism genes and metabolites were strongly enriched in both datasets fits with our current understanding of low-temperature stress resistance, as proline and arginine are specially used by mitochondria as a metabolic fuel and/or are involved as cryoprotectant or osmotic protectant of insect tissues under low-temperature exposure [35–37]. These findings suggest that this pathway plays an important role in improving the survival of flies under low-temperature stress.

## Gut bacteria aids the host to stimulate gene expression levels of the arginine and proline metabolism pathway during low-temperature stress

To confirm the response of the arginine and proline metabolic pathway to low-temperature stress, we used qPCR to assess the changes in the expression profiles of five important genes chosen on the basis of their roles in the development and survival of insect hosts under stressful conditions [38, 39]. The results showed that *K. michiganensis* reinfection strikingly induced the expression of argininosuccinate synthase (ASS), a precursor enzyme of arginine synthesis, by 14.70-fold 3 dpe to 10˚C, with a peak of 194.51-fold increase compared with that in ABX flies 5 dpe to 10˚C (Fig 5). *K. michiganensis* reinfection also significantly enhanced the mRNA levels of pyrroline-5-carboxylate reductase (Pro-C) and ornithine aminotransferase (Oat), which are required for the synthesis of proline, by 2.52- and 14.01-fold compared with the levels in ABX flies, respectively, 5 dpe to 10˚C. Consistent with these observations, increased levels of arginine and proline were found in the hemolymph of *K. michiganensis*-reinfected flies than those in ABX flies. Similarly, 5 dpe to 10˚C, the mRNA levels of other enzymes were also significantly enhanced in *K. michiganensis*-reinfected flies than that of ABX counterparts, e.g., arginase (Arg), required for catabolism of arginine to ornithine by 8.71-fold, and ornithine decarboxylase (ODC), required for the synthesis of putrescine by 5.26-fold, a likely reflection of the increased low-temperature resistance of the host induced by the gut bacteria. However, before the flies were exposed to low-temperature (0 dpe), the expression levels of these genes were not significantly different among different treatments, with only Oat expression being significant at 0 dpe, suggesting that in the presence of gut bacteria such as *K. michiganensis*, the host stimulates these genes only following exposure to low-temperature stress (Fig 5). Importantly, there were no significant differences in the transcript levels of these genes between conventional and *K. michiganensis*-reinfected flies before or after exposure to 10˚C, suggesting that *K. michiganensis* reinfection of ABX flies could upregulate the levels of these genes to the levels found in conventional flies. Taken together, our data demonstrate that *K.*

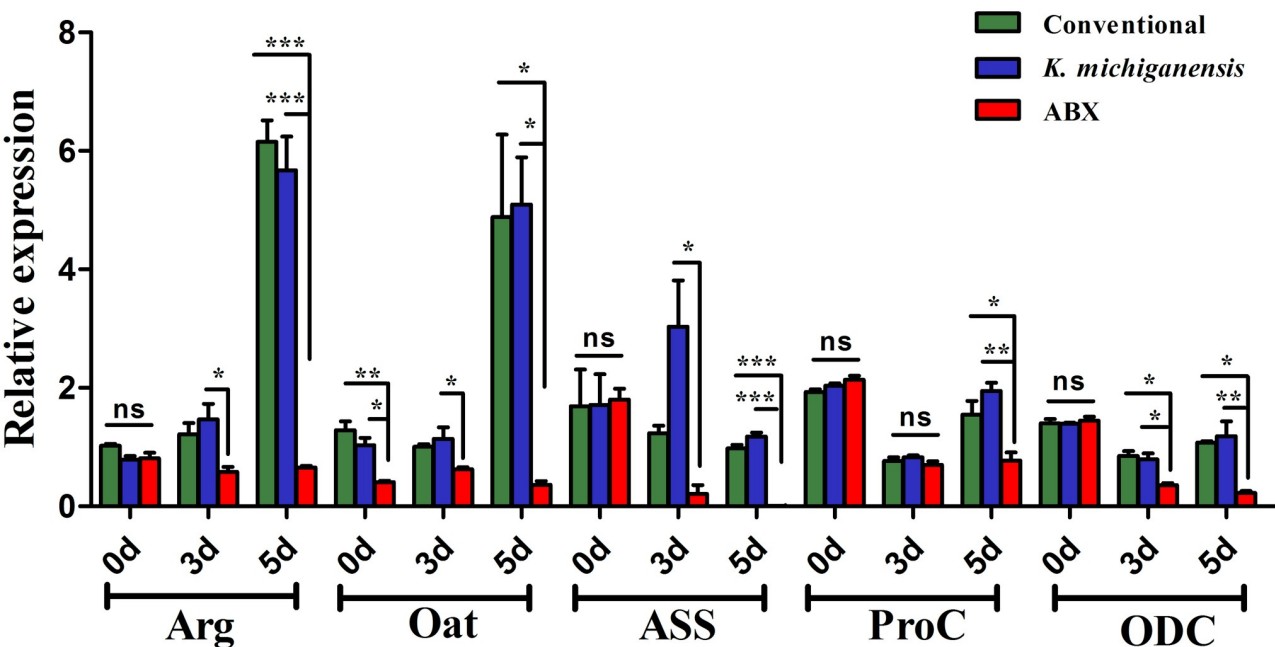

**Fig 5. Gut microbiota helps the host to elevate gene expression levels of the arginine and proline metabolism pathway postexposure to low-temperature stress.** qPCR analysis to assess relative expression levels of genes known to be required for the arginine and proline metabolism at 0 day, and stimulation after 3 days and 5 days postexposure to 10˚C in conventional, *K. michiganensis*-reinfected and ABX flies. One-way analysis of variance (ANOVA) and Tukey's test were performed. Statistical significance was indicated as follows: ***, $P<0.0001$; **, $P<0.001$; *, $P<0.05$; $P>0.05$ denoted as nonsignificant (ns). * Arginase (Arg), Ornithine aminotransferase (Oat), Argininosuccinate Synthase (ASS), Pyrroline-5-carboxylate reductase (Pro-C), Ornithine decarboxylase (ODC). Y-axis denotes time: 0 days (**27 ± 1˚C** or before exposure to 10˚C), 3 days and 5 days postexposure to 10˚C.

*michiganensis* reinfection helps the host to significantly stimulate the expression of these genes, which might account for enhanced low-temperature resistance.

## RNAi mediated silencing of arginine and proline genes reduce the survival of conventional flies under low-temperature stress

To validate the role of arginine and proline genes on the survival of flies under low-temperature stress treatment, we silenced four key genes (ASS, Pro-C, Arg and ODC) by injection of doublestranded RNA (dsRNA) to conventional flies. The dsRNA injection significantly reduced the mRNA levels of all four genes (Fig 6). The dsPro-C and dsASS silencing significantly decreased the Pro-C and ASS mRNA levels by 54% and 50% (Fig 6A and 6B) in conventional flies, respectively. After silencing the Pro-C and ASS genes, flies were subjected to low-temperature stress of 10˚C and their median survivals were significantly reduced by 57% and 36%, respectively (Fig 6E). The dsODC and dsArg treatment also significantly reduced the mRNA levels of ODC and Arg of conventional flies (Fig 6C and 6D) but their median survivals were only reduced by 14% and 7%, respectively (Fig 6E). These data suggest that ASS and Pro-C, which are required for the synthesis of arginine and proline, are important for the survival of flies under a low-temperature stress environment.

## Combined effect of key genes silencing on the survival of conventional flies postexposure to low-temperature stress

According to the previous experiments, we found that two key genes Pro-C and ASS silencing considerably reduced the median survivals of flies (Fig 6E). Next, we made the combination of

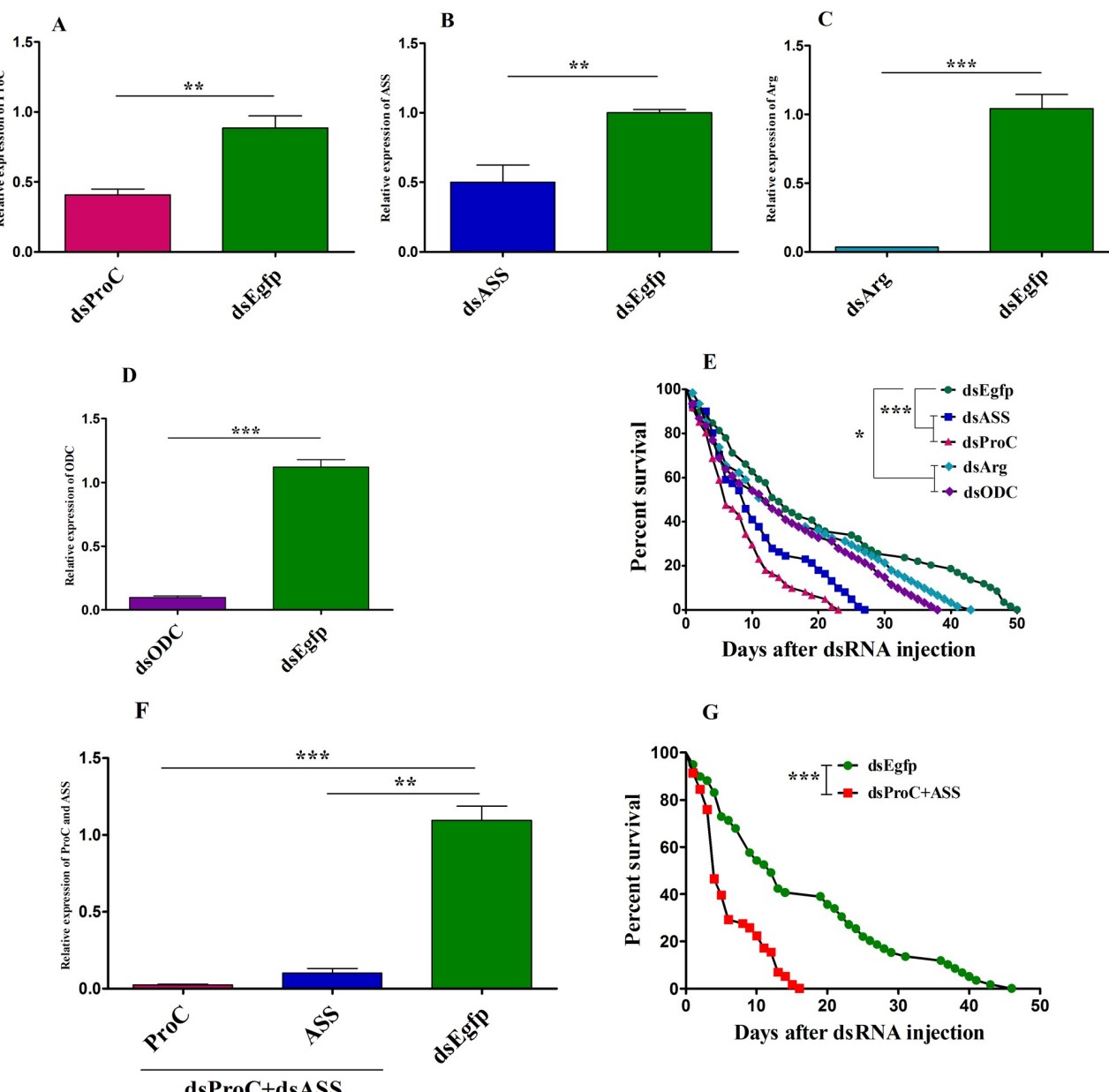

**Fig 6. Arginine and proline gene silencing via RNAi reduces the conventional flies survival time under low-temperature stress.** Gene silencing efficiency in conventional *B. dorsalis* (n = 10 in each repeat) injected with dsEgfp as control treatment or dsPro-C (A), dsASS (B), dsArg (C), dsODC (D). Survival curves of conventional flies (n = 59–61 in each condition) injected with dsEgfp or dsPro-C, dsASS, dsArg, dsODC (E) postexposure to 10˚C. Combined effect of dsPro-C + dsASS on gene silencing efficiency in conventional flies (n = 10 in each repeat), control group was injected with dsEgfp (F). Survival curves of conventional flies postexposure to low-temperature stress (n = 58–59 in each condition) after dsPro-C + dsASS injection (G). Experiments were performed in three biological replicates. Error bars indicate mean with SE. Significant difference was determined by Student's t test at $P<0.05$ in dsRNA treatment. Survival curves of conventional flies were analyzed by log-rank analysis at $P<0.05$.

dsPro-C and dsASS to silence these genes and evaluated the synergistic effect on the survival of conventional flies under low-temperature stress. The results showed that injection of dsPro-C + dsASS significantly decreased the Pro-C and ASS mRNA levels in conventional flies (Fig 6F), and as a result their median survival was markedly reduced by 67% postexposure to low-temperature stress of 10˚C (Fig 6G). These results suggest that a combined decrease in Pro-C

and ASS mRNA levels might cause a highly significant reduction in the survival time of flies during low-temperature stress.

### Functional validation of arginine and proline for the improvement in low-temperature stress resistance

Finally, we tested whether injection of the main constituents of the arginine and proline metabolism pathway could play their role as cryoprotectants to improve the survival of ABX flies at 10˚C. The results showed that the median survival times of ABX flies were significantly increased after individual microinjection of L-arginine (10 days) and L-proline (11 days) compared with that of the ABX control (6 days) injected with water, representing 67% and 83% increase, respectively (Fig 7), confirming their role in increasing the host's low-temperature stress resistance. Furthermore, we evaluated the synergistic application of L-arginine + L-proline to the ABX flies' survival. The results showed that after injecting the combination of L-arginine + L-proline, the median survival time of ABX flies was considerably increased by 150% (15 days) compared with the ABX control (Fig 7), suggesting the synergistic potential of these amino compounds under cold stress. Despite being the upregulation of fatty acid molecules or enzymes in both 'omics' approaches, we did not find the improvement in host low-temperature stress resistance after microinjection of saturated or unsaturated fatty acids (S4A and S4B Fig). To understand whether fat synthesis or storage consumptions are also responsible for the survival mechanisms, TG levels were assessed in the fly fat body tissue among conventional, *K. michiganensis*-reinfected or ABX treatments. However, we did not find significant differences in the TG levels among those different

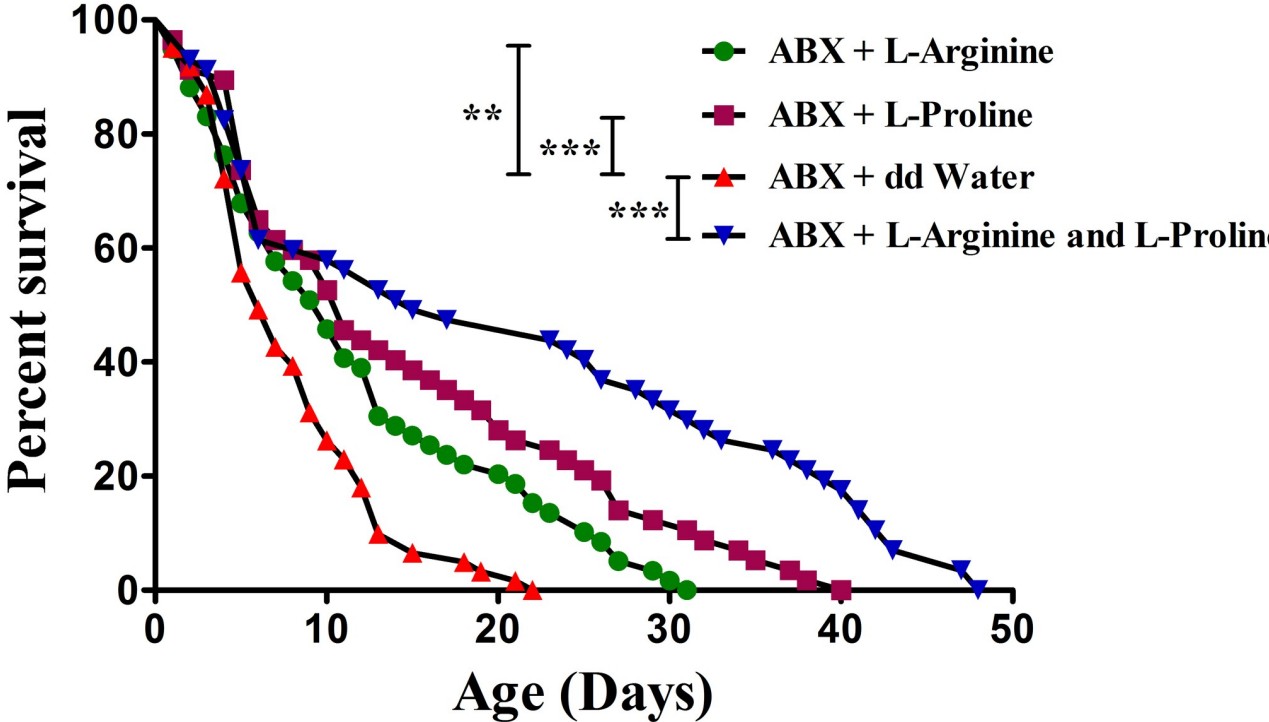

**Fig 7. The median survival times of ABX flies significantly increased after microinjection of L-arginine and L-proline (individually) and L-arginine + L-proline (in combination).** Survival curves under low-temperature stress of 10˚C for ABX flies injected with L-arginine (*P* = 0.0018), L-proline (*P*<0.0001), L-arginine + L-proline (*P*<0.0001) or double-distilled water (n = 57–61 for each condition, log-rank test, ** *P*<0.005; *** *P*<0.0001).

treatments (*P*>0.05, S4C Fig), indicating that fatty acid molecules might not be involved in the host resistance to cold stress.

## Gut bacteria maintains mitochondrial morphology and ATP levels during low-temperature stress

To investigate a link between gut bacteria and mitochondria activity for the arginine and proline induction to promote host resistance during low-temperature stress, we observed gut tissues of conventional, *K. michiganensis*-reinfected or ABX flies 5 dpe to 10˚C, through transmission electron microscopy (TEM). The results showed that in the absence of gut bacteria (ABX flies), low-temperature stress caused severe destruction in cristae structure and thus resulted in abnormal circular shapes of mitochondria compared with conventional flies (Fig 8A–8C). On the other hand, *K. michiganensis* inoculation helped the host to maintain mitochondrial morphology, clearly different from that of ABX flies following exposure to low-temperature stress. Furthermore, mitochondrial morphology of ABX flies were severely distorted compared with conventional or *K. michiganensis* inoculated flies (Fig 8D; percentage of distorted mitochondria in Conventional: 5.027 ± 0.9992%; *K. michiganensis*: 14.95 ± 1.709%; ABX flies: 77.99 ± 1.838%, *P*<0.0001). Number of cristae in ABX flies were also significantly reduced by 71% than that of *K. michiganensis* fed flies (Fig 8E). These experimental observations led us to conclude that, once inoculated, *K. michiganensis* might help the ABX flies to maintain a dynamic equilibrium in mitochondria and improves its metabolic functions that might induce arginine and proline to the levels sufficient to promote host resistance to low-temperature stress.

Moreover, to define the mitochondria functionality and metabolic mechanisms that promote host survival during cold stress, we measured the amount of ATP in the gut tissues of these flies. We found that ATP concentration was significantly reduced by 78.64% in the gut tissues of ABX flies compared with the conventional flies 5 dpe to 10˚C (Fig 8F, *P*<0.05). However, after the feeding of live *K. michiganensis* to ABX flies, ATP concentration in the gut tissues was reestablished and significantly increased by 377% compared with those in the ABX flies (Fig 8F).

## Discussion

Gut microbiota provides a wide range of beneficial effects for their hosts, including vitamins, energy metabolism, reproduction, immune function, and longer lifespan in unfavorable environments [21, 24, 25, 40]. In our previous study, we found that Enterobacteriaceae is the prevalent community in the digestive tract of *B. dorsalis* [31]. In particular, *Klebsiella* was identified as the dominant symbiotic bacterial genus in the *B. dorsalis* gut [24], suggesting that those bacterial symbionts may contribute to the biology of the host fly species. Pieterse and colleagues found that *B. dorsalis* adults showed a slight but statistically significant expansion in survival after exposure to a low hardening temperature of 10˚C [41], but the role of gut symbionts on the host biology during low-temperature stress has not previously been described. In this study, we found that the gut microbiota of *B. dorsalis* functions to promote host resistance to temperature stress, as gut bacteria-deprived flies do not live as long as conventional flies at 10˚C.

Furthermore, through the recolonization of ABX flies with dominant members of *B. dorsalis* indigenous gut microbiota, a key gut symbiont, *K. michiganensis* BD177, was found to be capable of promoting the survival time of the ABX flies to the level of the conventional flies during low-temperature stress of 10˚C. Our data also supports the idea that only *K. michiganensis* can form a sustained colony in the ABX flies gut equivalent to the levels found in

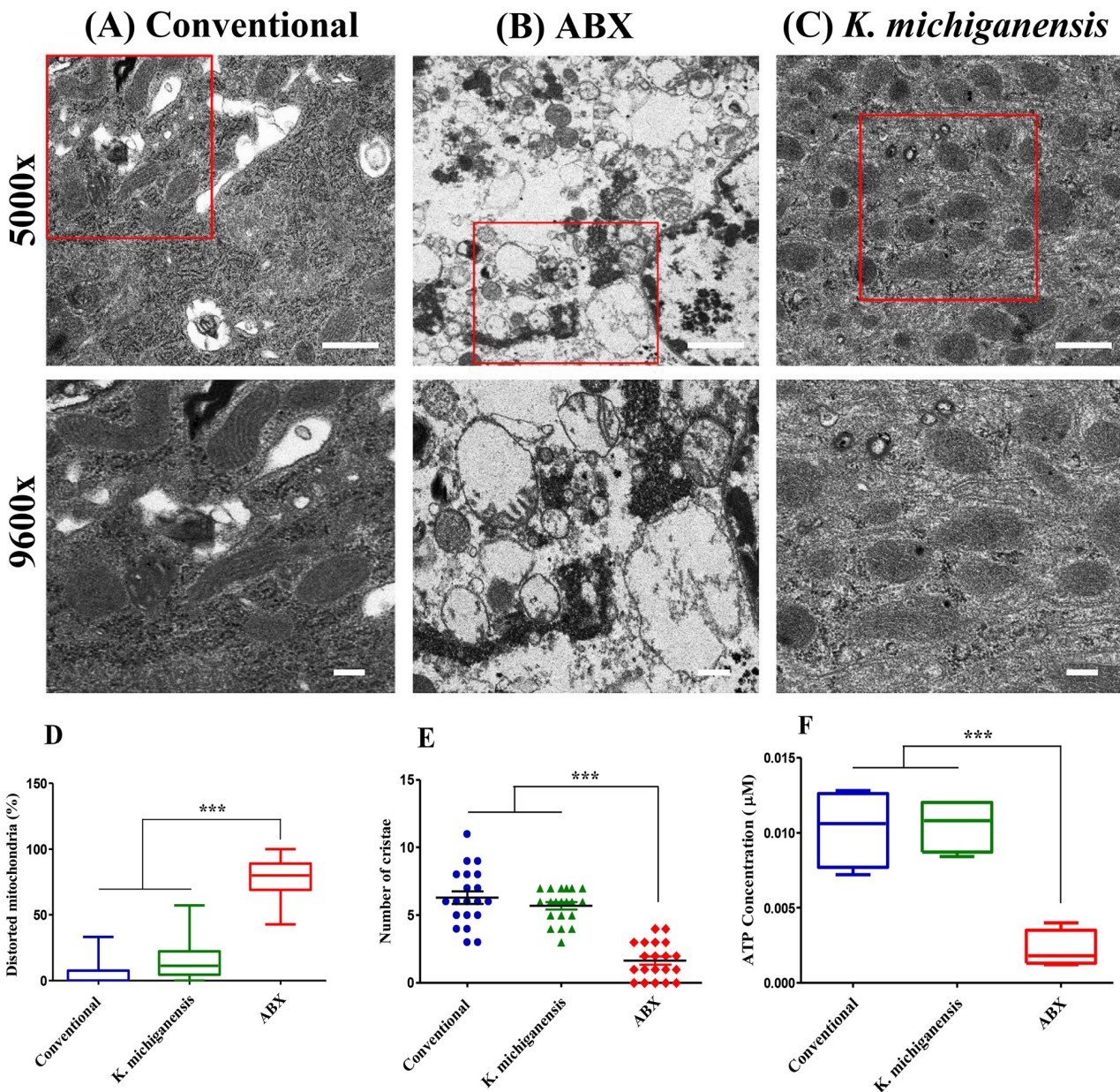

**Fig 8. Gut bacteria maintains mitochondrial morphology and ATP levels during low-temperature stress.** Representative TEM images of gut tissues of conventional (A), ABX (B), *K. michiganensis* (C) reinfected flies 5 dpe to 10˚C. The boxed areas were enlarged in the bottom panels. Scale bars, 1 µm (5000×, top) and 500 nm (9600×, bottom). (D) Percentage of distorted mitochondria in gut tissues of conventional, ABX, *K. michiganensis* reinfected flies; percent distorted mitochondria observed in Conventional were 5.027 ± 0.9992%; *K. michiganensis*: 14.95 ± 1.709% and ABX flies: 77.99 ± 1.838%, $P<0.0001$. (E) Comparison of the number of mitochondrial cristae in gut tissues of conventional, ABX, *K. michiganensis* reinfected flies n = 20. (F) Comparison of ATP levels in the conventional, ABX, *K. michiganensis* reinfected flies gut tissues 5 dpe to 10˚C n = 4. One-way analysis of variance (ANOVA) and Tukey's test were performed ($P<0.05$; and not significant denoted by ns, $P>0.05$) for the analysis of D, E and F figures.

conventional flies and/or significantly higher than that of all other tested strain, suggesting that *K. michiganensis* quantity is influential to promote host resistance ability to low-temperature stress. In future, it may be interesting to determine if microbial quantity impacts fly behavior, physiology, or survival during stressed environment, particularly under multimicrobial experimental conditions. The results of this study are in accordance with the

findings of previous studies. For example, it has been shown that stress caused by irradiation significantly reduced the levels of the main gut flora 'Enterobacteriaceae' that resulted in the decline of various aspects of ecological fitness in sterile *Ceratitis capitata* [26] and *B. dorsalis* [32]. However, the addition of the gut symbiont *K. oxytoca* to the postirradiation diet restored the survival and competitiveness of sterile *C. capitata* males and resulted in decreased levels of potentially pathogenic *Pseudomonas* sp. [26]. In our previous study, we found that gut-symbiont probiotics *K. oxytoca* reversed the radiation-induced host fitness decline, including survival time of *B. dorsalis* [32]. However, our recent genome wide sequencing and phylogenetic analysis showed that the *Klebsiella* sp. BD177, which was isolated from the *B. dorsalis* gut and previously identified as *K. oxytoca* on the basis of 16S RNA sequencing [32], has 97.74% ANIm with the reference genome of *K. michiganensis* DSM 25444 (= W14) (Accession: PRJNA388837) but only 92.20% ANIm with the *K. oxytoca* ATCC 13182 reference genome; thus, it had been classified as *K. michiganensis* BD177 (Accession: PRJNA602959), which was separated and independent from *K. oxytoca* according to average nucleotide identity (ANI), a genomic gold standard for bacteria species identification [42]. After cleaning the gut bacteria of *Drosophila melanogaster*, reinfection with a live gut symbiont *Issatchenkia orientalis* has also been demonstrated to promote the lifespan of nutritionally stressed flies [43]. More recent studies show that HK gut microbes can promote the lifespan of nutritionally deprived axenic *Drosophila* in repeated and dose-dependent manner at normal room temperature [43–45]. In contrast, our study show that increasing the quantity of HK *K. michiganensis* (recurring) did not promote the ABX flies survival to the levels of live *K. michiganensis* inoculated or conventional flies, suggesting that gut microbes may act as a source to enhance host metabolism, rather than providing direct nutritional benefits following exposure to low-temperature stress. This contradiction might be due to the different environmental conditions, e.g., different temperature conditions. In future research, the use of HK microbe supplementation may help fully describe microbial influences, i.e., to identify a direct nutritional contribution of microbes to the host under different stressed environmental conditions.

To the best of our knowledge, few studies have employed the symbiotic role of *Klebsiella* genus strains to explore the host fitness under stressed conditions. For example, Behar and colleagues suggested that inoculation of *K. oxytoca* to antibiotic-treated medflies contributes to the host nitrogen and carbon metabolism, copulatory success and development [46]. The gut symbiotic species *K. oxytoca* has also been reported to encode for the nitrogen fixation pathway by converting N2 to ammonia [47].

The gut microbiota not only provides essential nutrients to the hosts [20, 21] but also aid insect resistance to environmental stress via regulating host-signaling pathways and metabolism [24, 48]. An example of *Drosophila* host-microbe interaction is that the activity of pyrroloquinoline quinone-dependent alcohol dehydrogenase (PQQ-ADH) of a commensal bacterium, *Acetobacter pomorum*, stimulates the host insulin/insulin-like growth factor signaling (IIS) pathway to regulate homeostasis programs monitoring intestinal stem cell activity, body size, energy metabolism and development rate [48]. Similarly, after deprivation of a key gut symbiont '*Citrobacter* sp.' via antibiotic treatment in a resistant *B. dorsalis* strain, the host showed a decreased level of trichlorfon resistance [24]. Furthermore, recolonization of the susceptible *B. dorsalis* strain with '*Citrobacter* sp.' increased the host resistance by degrading the highly toxic trichlorfon insecticide into less toxic chloral hydrate and dimethyl phosphite compounds and possibly via stimulation of organophosphorus hydrolase (OPH-like) genes [24].

By combining transcriptomic and metabolomic approaches, we found that gut symbionts, particularly *K. michiganensis* BD177, help the host *B. dorsalis* upregulate the levels of 'cryoprotectant' transcripts and metabolites, which increases its resistance to long-term low-temperature stress by stimulating the host arginine and proline metabolism pathway. The accumulation of

small cytoprotectant molecules is one of the hallmarks in insect adaptation to low-temperature [49]. In insect low-temperature stress resistance, these molecules, also known as 'cryoprotectants' are mostly sugars, polyols [50] and free AAs [51]. Elevated levels of arginine, proline, ornithine and putrescine in nonABX flies, especially in ABX flies reinfected by the key gut symbiont *K. michiganensis*, suggest that the presence of gut microbiota can stimulate the host metabolism to produce these cryoprotectants, which, in turn, promotes host survival during low-temperature stress.

In fact, other studies have shown that arginine and proline metabolic products are important for the growth and development of *Drosophila* larvae and adults under a cold stress environment [35, 36, 49]. Misener et al. suggested that proline has a special role in dipterans, accumulating early in cold response and later may provide important energy reserves to maintain ATP levels [35]. In the current study, we found that RNAi mediated silencing of two key genes, i.e., Pro-C and ASS, which are precursor enzymes required for proline and arginine synthesis, significantly reduced the survival time of conventional flies following exposure to low-temperature stress. Combined effect of dsPro-C + dsASS further confirmed that median survival time of conventional flies was considerably reduced compared with that of control (dsEgfp) and individual dsRNA applications. These findings demonstrate that dsRNA targeting Pro-C or ASS may result in the low production of proline and arginine molecules which in turn decrease the survival time of flies. *Drosophila* transcript displayed a higher level of Pro-C expression in response to low-temperature stress, which shows that a large amount of proline is essential for *Drosophila* metabolism and survivorship during temperature stress [35]. Diet supplemented with proline conferred partial freezing tolerance by *D. melanogaster* larvae under winter acclimation [36]. Kostal and his colleagues also showed that proline and arginine-augmented diets can improve the freeze tolerance of *Drosophila* adults to 42.1% and 50.6%, respectively [49]. In agreement with previous studies, our data demonstrate that the individual or combined injections of L-proline and L-arginine to ABX flies can increase the survival time postexposure to low-temperature. Two possible mechanisms by which high concentrations of arginine and proline might stimulate low-temperature resistance could be as follows; first, proline could reduce the proteins partial unfolding and prevent membrane fusions during low temperature [49]. Second, proline and arginine both are unique among amino compounds in their capability to form supramolecular combinations that possibly bind partially unfolded proteins and cause hindrance in their aggregation under increasing freeze dehydration [49].

Arginine and proline synthesis, which requires mitochondrial activity, may suggest that mitochondrial activity can be maintained even at low temperature in insects [35–37]. Mitochondria are recycled in a dynamic equilibrium between opposing processes of fusion and fission. These processes are needed to control cell quality by substituting defective mitochondria with new and healthy mitochondria during high levels of cellular stress [52]. Herein, our results showed that, once inoculated, *K. michiganensis* aids the ABX flies in the maintenance of mitochondrial dynamic equilibrium that might restored both arginine and proline to the normal levels. Previous studies have also shown that cristae formations increase the surface area of the mitochondrial inner membrane and plays key roles in maintaining ATP production [53]. Any destruction to the cristae structure results in defective ATP production [54, 55]. In agreement with these findings, our data further confirmed that due to the destruction of mitochondrial cristae structure in ABX flies, the ATP concentration was significantly decreased than that of *K. michiganensis*-reinfected flies. However, we are currently unable to say how gut microbiota are able to maintain healthy mitochondria during cold stress. Thus, the connection between the gut bacteria and the mitochondria dynamic equilibrium, e.g., fission and fusion, during environmental stresses should be studied in the future. It might also be interesting to

investigate the mitochondrial morphology after the injection of highly expressed amino acids, e.g., L-proline and L-arginine and/or by silencing their corresponding genes in the future. Several antibiotics have been reported to be toxic to mitochondria in other animals [56]. It might be valuable to conduct cold stress studies using a truly axenic or germ-free flies, without using antibiotic treatments. We were unable to perform a low-temperature challenge on truly axenic or germ-free flies, however, our study still provides the clear effects of *K. michiganensis* reinfection that rescues the survival of ABX flies and helps the host to maintain mitochondrial morphology at low-temperature stress, suggesting that the reduced median survival time of ABX flies, which were generated via treatment with oral antibiotics cocktails as most researchers did in this research field [33], at low- temperature resulted from elimination of the gut microbiota, not from their toxic effects to mitochondria. Until this theory has been tested further, we can conclude that in the presence of gut microbes, these charged and polar amino acids, in particular the modulation of the host's arginine and proline metabolism via influencing mitochondrial functionality, are central to the acquisition of long-term low-temperature stress resistance.

## Conclusion

Host-microbe beneficial interactions of an organism encompass a large suite of molecular, physiological and biochemical adjustments under stressful environments [57]. It is this suborganismal or host-microbe 'tuning' of function that ultimately governs the fitness and survival of the organism during low-temperature stress. Our results show that gut microbiota of *B. dorsalis* promotes survival time and hemolymph total amino acid level during long-term low-temperature stress (10˚C). By combining the two 'omics' approaches, we have also provided strong support for the host acquisition of low-temperature stress resistance via the arginine and proline metabolism pathway, which is stimulated by gut bacteria symbiosis. Furthermore, we explored that gut microbes, particularly the *K. michiganensis*-host interaction, promote the mitochondrial functionality during low-temperature stress. These observations offer perspectives for future studies using the *Bactrocera* gut model for investigating host-microbe interactions in other animals.

## Materials and methods

### Fly culture

*B. dorsalis* were raised as described by Li et al. [58]. Briefly, flies collected from Guangzhou, China, have been reared for approximately 20 generations at the Institute of Urban and Horticultural Pests of Huazhong Agricultural University, Wuhan. Larvae were raised in bananas, and adult flies were kept in cages at 27 ± 1˚C under a photoperiod of 12 h light:12 h dark and fed with a sterile liquid diet (a mixture of 2.5% yeast extract, 7.5% sugar, 2.5% honey and 87.5% $H_2O$). The complete study design timeline is given in Fig 9.

### Initial experiments

**Isolation and identification of cultivable bacteria.**   To identify key symbiotic bacteria, cultivable gut bacteria of 5 dpe normally reared (28˚C) and low-temperature (10˚C)-exposed conventional fruit flies were isolated and identified by CFU assay. Briefly, different powers of serial dilutions of the 10 homogenized gut extracts, e.g., $10^{-5}$ and $10^{-7}$ [as described in 32], of those two different groups were made, and 100 µl of each dilution was inoculated in triplicate via the spread plate method on Luria-Bertani (LB) solid media under aerobic conditions. Then, plates were placed at 30˚C for 24 h for incubation. Subsequently, individual and

## Schematic representation of the experimental design

**Fig 9. Schematic representation of the study design.** The lines with arrowheads are the different bioassays conducted at different time points. The arrow head line ($\rightarrow$) on x-axis indicates that survival experiment continued until the mortality of all flies.

morphologically distinct colonies were inoculated into the corresponding liquid media and shaken at 200 rpm and 30°C overnight to produce DNA for extraction or for preservation in 60% sterile glycerol stocks that were kept at -80°C for further use. The bacterial DNA extraction was performed with a Hi-Pure Bacterial DNA Kit (Magen) following the protocol instructions and used for the amplification of 16S *rRNA* genes using the primer pair 27F and 1492R (S1A Table). Subsequently, we used a PCR purification kit (Axygen) to purify the ~1.4 kb PCR products and subjected the products to bidirectional Sanger sequencing and then BLASTed the sequence on EzBioCloud [59] for the identification of cultivable bacterial strains.

Similarly, to confirm whether the host is able to maintain an adequate supply of gut bacteria to benefit health throughout life postexposure to 10°C, we measured the CFUs of conventional, ABX flies and all other tested strains at two different time points 5 dpe and 10 dpe to 10°C. In short, $10^{-5}$ and $10^{-7}$ powers of serial dilution of the 10 homogenized gut extracts of those different treatment groups at two different time points were made, and 100 μl dilutions were inoculated in triplicate using the spread plate method on Luria-Bertani (LB) solid media under aerobic conditions and incubated at 30°C for 24 h. Next, CFUs resulting from the bacterial colonies on each plate were averaged and analyzed within and between samples.

**Antibiotic treatment.** Following an initial antibiotic sensitivity test, an antibiotic cocktail of streptomycin and penicillin with 5:3 μg/ml supplemented with sterile liquid diet was fed to newly eclosed flies for 5 consecutive days to generate ABX flies. To determine the efficacy of

the antibiotics, bacterial CFUs were determined by spread plating aliquots of 10 dissected guts homogenized in 1 ml double-distilled water. Gut bacterial loads were quantified by qPCR using total bacteria primer (uni331-F and uni771-R, S1A Table) on DNA samples obtained from the gut tissues according to the previously described method [60].

**Microinjection.** Highly expressed amino acids (AAs), e.g., L-arginine and L- proline (Sigma-Aldrich, USA) were dissolved in ddH$_2$O to a final concentrations of 8 g/L and 15 g/L, respectively, for individual or combined injections to ABX flies [61]. Saturated fatty acids, e.g., arachidonic acid and linoleic acid (Sigma-Aldrich, USA), dissolved in 0.1% dimethyl sulfoxide (DMSO) [62] were injected into ABX flies prior to low-temperature exposure. Furthermore, other types of fatty acid molecules present in our metabolomics data, e.g., Geraniol, Nona-decylic acid, 6-Keto-decanoylcarnitine, 3-Hydroxy-9-hexadecenoylcarnitine, 3-Hydroxy-9-hexadecenoylcarnitine, Latanoprost, Heptadecanal (Rhawn-Shanghai, China), Dodecanoic acid and Ricinoleic acid (Yuanye-Shanghai, China) were dissolved in 0.1% DMSO and also injected into the ABX flies before exposure to low-temperature stress treatment. ABX flies injected with water or DMSO were considered as controls. An injectMan N12 instrument (Eppendorf, Germany) equipped with a FemtoJet microinjection system was employed to perform microinjection. The glass capillaries were prepared using 50 μl glass micropipettes and a puller at heat level of 60.4˚C (PC-10, Narishige, Japan). The microinjection conditions were set to a Pi of 570 hpa and a Ti of 0.2 seconds. A volume of 200 nl solution was then injected into each fly [63].

## Fitness parameter tests

**Longevity and survivorship.** In trial 1, to determine the role of the gut symbiont in survival time improvement, ABX flies and conventionally reared adult flies (n = 60 individuals from each treatment) were placed into a 17 cm × 8 cm × 7 cm box containing liquid diet and subjected to low- to medium-temperature treatments (5˚C, 7.5˚C, 10˚C, 15˚C). Subsequently, 10˚C was screened for trial 2 experiments in which conventional, ABX control and ABX flies reinfected with all 15 isolated strains were subjected to 10˚C stress treatment. ABX controls were fed only with sterile liquid diet. ABX flies injected with L-arginine, L-proline, and other fatty acid metabolites were also subjected to the same low-temperature for the survival assays. Then, we placed the boxes in a growth chamber at 10˚C (RH: 60–70% and 12 h light: 12 h dark), and the number of dead flies was inspected at 24 h intervals. The incubator humidity and temperatures were monitored with humidity and temperature loggers. Sterile liquid diet was provided ad libitum till the end of survival or longevity assays. In addition, the median survival time was 5 days in ABX flies; therefore, we selected 5 dpe as the main timepoint for our subsequent studies, including RNA-sequencing and UPLC/MS analysis.

**Reinfection of ABX flies with gut symbionts.** In trial 2 fitness tests, to acquire the key bacterium among the all isolated strains, we reinfected the ABX flies with all 15 isolated strains (S1B Table). For monoculture experiments, equivalent levels of each microbe were introduced to ABX flies in order to ensure their potential beneficial effects (if any) on the survival of flies under low-temperature stress. The inoculum was prepared as follows: Optical density (OD) was measured on an overnight culture of each bacterial species used. Then, $10^{-5}$ and $10^{-7}$ powers of serial dilution of the bacterial cultures were made and 100 μl dilutions were inoculated on Luria-Bertani (LB) solid media using the spread plate method and incubated under aerobic conditions at 30˚C for 24 h. Next, we determined CFUs on each plate resulting from the different bacterial cultures, individually. The resuspension volume was determined based on the empirically determined constants (CFU ml$^{-1}$ at an OD of 2) to make an equal (normalized) dose of $10^8$ CFU/ml [46] for all tested strains. The cells from each bacterial cultures were

pelleted at 3600 X g for 15 minutes and resuspended in the liquid diet mixture of 2.5% yeast extract, 7.5% sugar, 2.5% honey and 87.5% $H_2O$ in equal parts and fed to the ABX flies for 5 days (Fig 9). HK gut bacterium (*K. michiganensis*) was prepared by autoclaving at 121˚C for 30 minutes, and aliquots were stored at -80˚C. A dose of $10^8$ CFU/ml HK *K. michiganensis* was provided to the ABX flies for 5 days (denoted as "single dose"). Flies reinfected with live or HK microbes (single dose) were then supplied with sterile liquid diet till the end of survival or longevity assays. Recurring supply of HK *K. michiganensis* to an initial dose of $10^8$ CFU/ml (equivalent to the initial inoculum of live *K. michiganensis*) or $10^9$ CFU/ml (increased dose) were supplied ad libitum till the end of survival or longevity assays. Conventionally reared adult flies were fed with sterile liquid diet throughout the experiment. We generally marked '0 days' before exposure to 10˚C, which was 10 days posteclosion, for all studies. Fifteen days post-eclosion or exposure to normal rearing temperature of 28˚C was also considered as 5 dpe, in the whole study to ensure the usage of same-aged flies (Fig 9).

**Triacylglyceride, amino acid and sugar measurement.** Adult fly hemolymph was collected 5 dpe to 10˚C using a centrifugation procedure as described previously [32] with slight modifications. Briefly, flies were immobilized on ice, and an incision was made across the pro-notum wall of the flies with a sterile microneedle that was prepared with a puller at 60.4˚C (PC-10, Narishige, Japan). Then, we placed ten flies (sex ratio 1:1) in a 0.5 ml microcentrifuge tube that was punctured at the bottom with a thumbtack. The 0.5 ml tube was inserted into another 1.5 ml microcentrifuge tube. Following centrifugation at 2000 rpm/min for 10 minutes, hemolymph was collected from the bottom of the outer 1.5 ml tube. The collected hemolymph was used immediately or stored at -80˚C for further use. Sugar and triacylglyceride (TG) were measured as previously described [64]. Amino acids were measured by using a total amino acid kit (Jiancheng, Nanjing, China) according to the manufacturer instructions. To assess the level of TG in the fly fat body, 14 flies (sex ratio 1:1) in each technical repeat were used for fat body tissue extraction 5 dpe to 10˚C. Four repeats were performed for each treatment. TG levels were measured by using a TG kit (Jiancheng, Nanjing, China) according to the manufacturer instructions.

**UV stress treatment.** Experimental irradiation in the growth chambers was provided by two daylight tubes (85 W, Philips Lighting, Markam, Ontario, Canada), and two UV tubes (30 W, 90 cm, Repti Glo 8.0, 33% UV-A and 8% UV-B, Rolf C. Hagen, Montre ´al, Quebec, Canada) for UV exposure. UV flux was measured using a UVX digital radiometer (UVP Inc., Upland, California, USA) equipped with specific probes for UV-A (peak at 365 nm) or UV-B (peak at 310 nm). Irradiance flux was measured at a distance of 35 cm under the light source, at the level of boxes containing experimental flies. To permit the passage of most UV-B and UV-A, while preventing exposure to UV-C (<280 nm), boxes were covered with a Saran Premium Wrap (S.C. Johnson and Son, Brantford, Ontario, Canada) absorbing short-wave radiation below 280 nm. Flies were exposed to UV radiation for 4 h daily. Between each 4 h UV irradiation interval, flies were returned to daylight irradiance only (equivalent to control setup). Survival rate of the test insects were monitored after every 24h interval (S5 Fig).

**ATP assay.** ATP concentration was measured by using ATP Assay Kit (Beyotime, cat. no. S0026) according to the manufacturer instructions. Briefly, guts of ten flies (sex ratio 1:1) 5 dpe to 10˚C were used to measure the ATP concentration. Four repeats were performed for each treatment. The luminescence was measured by Infinite F200 (Tecan, Swiss), and the results were compared to standards.

**UPLC/MS analysis.** Hemolymph collected from adult flies was diluted with acetonitrile (v/v, 50 μL: 200 μL) for protein removal [64]. Briefly, we used ten flies (sex ratio 1:1) in each repeat for hemolymph extraction from ABX, CK (conventional) and *K. michiganensis*-reinfected flies. Six repeats were used for each treatment. Then, we used ultra-performance liquid

chromatography (Waters Acquity BEH C18 column) coupled to a quadruple time-of-flight mass-spectrometer (Waters SYNAPT Q-TOF HDMS) for metabolomics profiling. The ion source was employed in positive (ESI+) electrospray ionization mode.

The original raw data from the mass spectrometer was converted to .mzXML format by Proteo Wizard. Peak alignment, retention time correction, and peak area extraction were performed using the XCMS program. Metabolites were identified using retention index and mass spectrum and were matched against the HMDB: http://www.hmdb.ca/ or METLIN databases. Differentially expressed metabolites were found by using the variable importance in the projection (VIP) value of the first principal component model, and the *P* values were calculated by Student's t-test. The threshold level to determine differentially expressed metabolites was set as VIP>1, FC>2.0 and *P* value<0.05. Volcano plots were used to infer the overall distribution of the DEMs. The peaks extracted from all experimental samples were subjected to PCA.

**RNA sequencing.**    RNA sequencing was performed in three libraries of ABX (ABX-1, ABX-2 and ABX-3), conventional (Conv-1, Conv-2, and Conv-3) and *K. michiganensis*-reinfected flies (*K. michiganensis*-1, *K. michiganensis*-2 and *K. michiganensis*-3). From each sample, 1.5 μg RNA was employed as input material for the preparation of RNA samples. Sequencing libraries were created using a NEBNext Ultra RNA Library Prep Kit for Illumina (NEB, USA) according to the manufacturer's instructions. Complete details of RNA sequencing analysis are available in "S1 Text" supporting informations. Briefly, differential expression analysis of two group comparisons (conventional vs. ABX and *K. michiganensis*-reinfected vs. ABX) was performed using the DESeq R package (1.10.1). Genes with an adjusted *P* value<0.05 and FC>2.0 found by DESeq were considered DEGs. We used KOBAS [65] software to examine the statistical enrichment of DEGs in KEGG pathways.

**dsRNA synthesis and delivery by injection.**    dsRNA was synthesized according to the previously described method [60]. Briefly, the target sequence fragments were amplified from the arginine and proline metabolism pathway genes (ASS, Pro-C, Arg and ODC) of *B. dorsalis* by nested PCR using template-specific primers conjugated with the T7 RNA polymerase promoter (5′-GGATCCTAATACGACTCACTATAGG-3′). The sequences of the primers are given in S1A Table. PCR product at 1 μg was used as the template for double-stranded RNA (dsRNA) synthesis using the T7 Ribomax Express RNAi System (Promega, Madison, WI, USA) according to the manufacturer's protocol. dsRNA was ethanol precipitated overnight, resuspended in RNase-free injection buffer (5mM KCl, 0.1mM sodium phosphate, pH 6.8) and quantitated at 260 nm using a NanoDrop 2000 Spectrophotometer (Thermo Fisher Scientific Inc.) before microinjection. The quality and integrity of dsRNA were determined by agarose gel electrophoresis. Microinjection was performed by using injectMan N12 instrument (Eppendorf, Germany) equipped with a FemtoJet microinjection system. The glass capillaries were prepared using 50 μl glass micropipettes and a puller at heat level of 60.4˚C (PC-10, Narishige, Japan). The injection condition was set to a Pi of 300 hpa and a Ti of 0.3 s. Gene silencing experiments were performed injecting 1 μl of a 2μg μl$^{-1}$ solution of dsRNA into the ventral abdomen of each fly (1–2 days old). 1 μl of dsRNA combination of dsPro-C + dsASS was also injected at the final concentration of 2μg μl$^{-1}$. Control flies were injected with dsEGFP.

**Real-time PCR.**    For gene expression analysis, 10 flies (sex ratio 1:1) were collected for RNA extraction at different time intervals (0, 3 and 5 dpe). An RNAiso Plus reagent (Takara, Japan) was used for RNA extraction. cDNA synthesis was performed using 500 ng total RNA with the Transcript RT Master Mix (Takara, Japan) according to the manufacturer's guidelines. RT-qPCR was performed (complete details are available in supporting informations S1 Text) on a BioRad MyIQ2 instrument using BioRad SYBR Green qPCR mix (BioRad, USA). All samples were analyzed in triplicate (technical repeats). Additionally, the efficiency of dsRNA-mediated gene silencing was also determined by collecting 10 flies (sex ratio 1:1) after

2 days post-injection. Three technical repeats were used and RT-qPCR was performed to assess the efficacy of dsRNA injection.

The loads of total bacteria were quantified by RT-qPCR using the 16S RNA gene-specific primers (S1A Table), and data were normalized for the host β-actin gene according to the previously described method [66]. Primer pairs used in RT-qPCR analysis are given in S1A Table. The qPCR data were analyzed using the $2-^{\Delta\Delta}CT$ method [67].

**Transmission electron microscopy (TEM).** Fly gut tissues were fixed with 3% glutaraldehyde in 0.1Mcacodylate buffer (pH 7.2)-containing 0.1% CaCl2 for 2 hours at room temperature (RT). They were rinsed three times with 0.1Mcacodylate buffer at 4˚C. Next, they were postfixed with 1% OsO4 (Osmium tetroxide) in 0.1Mcacodylate buffer-containing 0.1% CaCl2 for 2 hours at 4˚C. After rinsing with cold distilled water, the tissues were dehydrated slowly with an ethanol series and propylene oxide at 4˚C. The samples were embedded in Embed-812 (EMS, PA). After polymerization of the resin at 60˚C for 36 hours, serial sections were performed with a diamond knife on an ULTRACUT UC7 ultramicrotome (Leica, Germany) and mounted on formvar-coated slot grids. Sections were stained with 4% uranyl acetate for 30 minutes and lead citrate for 10 minutes. They were observed using a Tecnai G2 Spirit Twin transmission electron microscope (FEL, USA).

**Statistical analysis.** All the results for different groups were analyzed using Student's t-test or a one-way analysis of variance (ANOVA) and Tukey's test using SPSS 20 (IBM Corporation, USA). CFU data of all bacterial strains were compared with live *K. michiganensis*-reinfected flies CFUs using ANOVA (Dunnett's multiple comparison test at **$P<0.05$, ***$P<0.005$, ns $P>0.05$). Survival statistics were calculated using a log-rank analysis and the Gehan-Breslow-Wilcoxon test. Graphs were made using Graph-Pad Prism 6.0 (Graph-Pad Software, La Jolla, CA, USA).

Distorted and healthy mitochondria taken from TEM gut images of different treatments were counted by using analyses toolbox of Photoshop (Adobe Inc., San Jose, CA, USA). Counting was made by at least two independent people blinded to the type of the samples. For each quantification at least 6 flies were used for each treatment group. Statistical analyses were based on the average numbers for each fly, and not based on total individual number of TEM gut images or organelle (mitochondria). When counting mitochondria present inside the gut cell images, sampling was not performed. All mitochondria that are present in the cell images were calculated without bias or elimination. Ten randomly selected cell images of gut were analyzed per fly per treatment group and counting was performed for all mitochondria present in each cell image. Although the total number of mitochondria varies from cell to cell, the percentage of distorted mitochondria was calculated per cell image and their averages determined the quantitative measure for each fly, which are displayed in the box & whiskers graphs in Fig 8D. The total number of mitochondria calculated and the total number of images analyzed for quantification of distorted mitochondria percentage are presented in 'table 2' in S1 Text (supporting methods).

The numerical data used in all figures are included in S3 Data.

## Supporting information

**S1 Fig. Survival curves of ABX and conventional flies at normal (28˚C) and different long-term low- to mild-temperatures stresses of 5˚C, 7.5˚C and 15˚C.** (A) There was a nonsignificant difference in the survival time of ABX and conventional flies under normal rearing temperature (28˚C). (B) A nonsignificant difference was observed between median survival time of ABX (2 days) and conventional (2 days) flies at 5˚C. (C) Under 7.5˚C temperature stress, there was a small but statistically significant percent survival change, but median survival time was not significantly different between ABX (5 days) and conventional (4.5 days) flies. (D)

Conventional fly median survival time was enhanced at 15°C (55 days) compared with that of ABX flies (10 days). Newly eclosed flies were treated with antibiotic solution for 5 consecutive days to generate ABX flies, while conventional flies were only fed sterile liquid diet. After 5 days posteclosion, both group of flies were fed with sterile liquid diet until mortality. n = 60 for each condition, log-rank (Mantel-Cox) test, *** $P<0.0001$, * $P<0.05$ and ns $P>0.05$). (TIF)

**S2 Fig. Survival curves of ABX flies recolonized by all isolated strains (Trial 2) and their microbial load in gut homogenates.** (A)The median survival time was extended in ABX flies recolonized with the following gut symbionts ($P<0.05$): *K. michiganensis* (13 days), *E. soli* and *C. koseri* (11 days), *E. tabaci* (9 days), *E. hormaechei* (8.5 days), *A. radioresistens* and *P. alcalifaciens* (8 days), *Leclercia adecarboxylata* (7.5 days), *Enterococcus faecium* and *P. vermicola* (7 days), *P. rettgeri* (6.5 days) and *A. bereziniae* (6 days) compared with 5 days in the ABX control. In contrast, median survival time was unaffected ($P>0.05$) after the recolonization of ABX flies with *Lactococcus garvieae*, *Serratia marcescens* and *Kluyvera ascorbata* compared with 5 days in the ABX control. Note: There were significant differences ($P<0.05$) in the median survival time between conventional flies and all other tested strains, except in conventional vs. *K. michiganensis*-reinfected flies ($P = 0.1275$), under low-temperature stress of 10°C. Log-rank (Mantel-Cox) test was used to compare the survival between different treatments. *A+ strain name (ABX flies recolonized by the above gut bacterial strains). (B and C) Microbial load in gut homogenates is maintained at different ages after feeding live *K. michiganensis* to ABX flies and in conventional flies. (B) Colony-forming units (CFUs) of flies after 5 dpe to 10°C; the average number of cultivated microbial communities resulting from CFU in ABX control fed with sterile liquid diet was $1 \times 10^5 \pm 5.7 \times 10^4$ CFUs gut$^{-1}$ (mean ± SE of 10 individual flies), representing 99.49% decreases vs. that of live *K. michiganensis*-reinfected flies, which showed $1.97 \times 10^7 \pm 1.87 \times 10^6$ CFUs gut$^{-1}$ (mean ± SE of 10 individual flies) ($P<0.0001$). No significant difference was observed in the average number of cultivated microbial communities between conventional ($2.103 \times 10^7 \pm 1.637 \times 10^6$ CFUs gut$^{-1}$) and live *K. michiganensis*-reinfected ($1.97 \times 10^7 \pm 1.87 \times 10^6$ CFUs gut$^{-1}$) ($P>0.05$) flies at 5 dpe to 10°C. The average numbers of cultivated microbial communities resulting from CFUs after 5 dpe to 10°C in ABX flies fed with *E. soli* were ($1.373 \times 10^7 \pm 7.965 \times 10^5$), *C. koseri* ($1.407 \times 10^7 \pm 6.96 \times 10^5$), *E. tabaci* ($1.053 \times 10^7 \pm 1.562 \times 10^6$), *E. hormaechei* ($1.203 \times 10^7 \pm 6.566 \times 10^5$), *A. radioresistens* ($9.267 \times 10^6 \pm 8.413 \times 10^5$), *P. alcalifaciens* ($8.867 \times 10^6 \pm 1.78 \times 10^6$), *Leclercia adecarboxylata* ($8.533 \times 10^6 \pm 4.256 \times 10^5$), *Enterococcus faecium* ($4.867 \times 10^6 \pm 4.256 \times 10^5$), *P. vermicola* ($7.9 \times 10^6 \pm 1.96 \times 10^6$), *P. rettgeri* ($4.767 \times 10^6 \pm 2.404 \times 10^5$), *A. bereziniae* ($3.7 \times 10^6 \pm 9.713 \times 10^5$), *L. garvieae* ($1.20 \times 10^6 \pm 1.528 \times 10^5$), *S. marcescens* ($1.367 \times 10^6 \pm 5.487 \times 10^5$) and *K. ascorbate* ($1 \times 10^6 \pm 2.082 \times 10^5$) CFUs gut$^{-1}$, and decreased by 29%, 27%, 45%, 37%, 52%, 54%, 55%, 74%, 59%, 75%, 80%, 93%, 92%, 94%, respectively, compared with CFUs obtained from live *K. michiganensis*-reinfected flies ($1.97 \times 10^7 \pm 1.87 \times 10^6$ CFUs gut$^{-1}$). (C) CFUs of flies after 10 dpe to 10°C; the average numbers of cultivated microbial communities resulting from CFUs in ABX control fed with sterile liquid diet was $7.667 \times 10^5 \pm 2.028 \times 10^5$ CFUs gut$^{-1}$, showing 96% decreases compared with live *K. michiganensis*-reinfected flies, which had populations of $1.937 \times 10^7 \pm 3.014 \times 10^6$ CFUs gut$^{-1}$. No significant difference was observed in the average number of cultivated microbial communities between conventional ($1.98 \times 10^7 \pm 2.38 \times 10^6$ CFUs gut$^{-1}$) and live *K. michiganensis*-reinfected flies ($1.937 \times 10^7 \pm 3.014 \times 10^6$ CFUs gut$^{-1}$) ($P>0.05$) at 10 dpe to 10°C. Importantly, CFUs of flies after 10 dpe to 10°C; the average numbers of cultivated microbial communities resulting from CFUs in ABX flies fed with *E. soli* ($1 \times 10^7 \pm 4 \times 10^5$), *C. koseri* ($1.04 \times 10^7 \pm 4.09 \times 10^5$), *E. tabaci* ($7.867 \times 10^6 \pm 7.796 \times 10^5$), *E. hormaechei* ($7.033 \times 10^6 \pm 7.513 \times 10^5$), *A. radioresistens* ($4.3 \times 10^6 \pm 1.255 \times 10^6$), *P. alcalifaciens* ($5.9 \times 10^6 \pm 2.042 \times 10^6$), *Leclercia adecarboxylata* ($7.533 \times 10^6 \pm 1.499 \times 10^6$), *Enterococcus faecium*

$(4.1 \times 10^6 \pm 3.215 \times 10^5)$, *P. vermicola* $(6.9 \times 10^6 \pm 1.96 \times 10^6)$, *P. rettgeri* $(3.767 \times 10^6 \pm 7.688 \times 10^5)$, *A. bereziniae* $(2.367 \times 10^6 \pm 1.12 \times 10^6)$, *L. garvieae* $(8.667 \times 10^5 \pm 1.856 \times 10^5)$, *S. marcescens* $(1.1 \times 10^6 \pm 4.163 \times 10^5)$ and *K. ascorbate* $(7.667 \times 10^5 \pm 1.764 \times 10^5)$ CFUs gut$^{-1}$, shown 48%, 46%, 59%, 63%, 77%, 69%, 61%, 78%, 64%, 80%, 87%, 95%, 94%, 96%, decrease respectively, compared with live *K. michiganensis*-reinfected flies, which had populations of $1.937 \times 10^7 \pm 3.014 \times 10^6$ CFUs gut$^{-1}$. CFU data of all bacterial strains were compared with live *K. michiganensis*-reinfected flies CFUs using ANOVA (Dunnett's multiple comparison test at $^{**}P<0.05$, $^{***}P<0.005$, ns $P>0.05$). The error bars indicate standard error of the mean (SEM) (n = 3). CFU represents the mean ± SE of 10 individual flies. (**D to F**) Gut bacterial load after 10 dpe and 20 dpe to 10˚C according to qPCR analysis on fly gut (n = 15) using total gut bacteria or species-specific primers. (D) Total gut bacteria, (E) *K. michiganensis* and (F) Enterobacteriaceae. Microbial load in gut homogenates was maintained after feeding live *K. michiganensis* and/or in conventional flies. Three biological repeats were conducted. Error bars indicate mean with SE. Significant difference determined by the Student's t-test at $^{**}P<0.005$, $^{***}P<0.0001$, ns $P>0.05$. Note: we could not quantify the microbial load of ABX flies reinfected with HK *K. michiganensis* (Single dose) or ABX control inoculated with liquid diet at 20 dpe to 10˚C because all flies reinfected with HK *K. michiganensis* (Single dose) or ABX control fed with liquid diet died before 20 dpe to 10˚C.
(TIF)

**S3 Fig. Validation of RNA-Seq data by real-time qPCR in conventional vs. ABX (A) and *K. michiganensis*-reinfected vs. ABX (B) groups.** Data are presented in Log2Fold Change. In the presence of gut microbes, some of these HSP, *ZFP* and *STK* transcripts were also downregulated in those comparison groups, while the transcripts related to arginine and proline metabolism were only upregulated in those group comparisons in response to low-temperature stress and overlapped in both omics approaches. Therefore, arginine and proline pathway was chosen to explore the low-temperature stress resistance. * Target of rapamycin genes (Eif4b, Rheb), protein processing in endoplasmic reticulum (cold shock protein e-1: csp e1; heat shock protein family: Hsf N, Hsf 23, Hsp90, Hsp70; serine/threonine kinase proteins: Stk32, Stk Tao; saccharopine dehydrogenase: Sdh; Vitellogenin-1: Vg-1).
(TIF)

**S4 Fig. Fatty acids did not improve survival of ABX flies during low-temperature stress.**
(A) Survival curves at low-temperature stress of 10˚C, microinjection of arachidonic acid $(P = 0.1005)$, linoleic acid $(P = 0.0916)$, Geraniol $(P = 0.9355)$, 6-Keto-decanoylcarnitine $(P = 0.9643)$, Dodecanoic acid $(P = 0.5675)$, Nonadecylic acid $(P = 0.2949)$, 3-Hydroxy-9--hexadecenoylcarnitine $(P = 0.9518)$, Latanoprost $(P = 0.0547)$, Heptadecanal $(P = 0.5722)$ and Ricinoleic acid $(P = 0.1754)$ to ABX flies compared with ABX control injected with DMSO (n = 60 for each condition, log-rank test, $P>0.05$ denoted as nonsignificant (ns). (B) FASN gene expression was not stimulated during low-temperature stress. The relative expression levels of fatty acid synthase (FASN) gene, a key enzyme that catalyzes the reductive synthesis of long-chain fatty acids (e.g., arachidonic acid, linoleic acid) at 0 day and 5 dpe to 10˚C in conventional, *K. michiganensis*-reinfected and ABX flies. One-way analysis of variance (ANOVA) and Tukey's test were performed. Statistical significance was indicated as follows: $P>0.05$ denoted as nonsignificant (ns). (C) TG levels were not significantly different in the fly fat body tissues among conventional, *K. michiganensis*-reinfected or ABX treatments (One-way analysis of variance (ANOVA) and Tukey's test were performed. Statistical significance was indicated as follows: $P>0.05$ denoted as nonsignificant (ns).
(TIF)

**S5 Fig. Gut symbionts does not promote survival under UV stress treatment.** The median survival time for conventional and ABX flies fed with *K. michiganensis* were 20.50 and 15 days, respectively, compared with 18.50 days found in the ABX flies. These results showed that there were non-significant differences in the median survival time among different treatments, suggesting that gut bacteria e.g., *K. michiganensis* might not have symbiotic effect to provide resistance against UV stress (n = 60 for each condition, log-rank test, *P*>0.05 denoted as nonsignificant (ns).
(TIF)

**S1 Data. a to f Metabolomics data.**
(XLSX)

**S2 Data. a to d Transcriptomics data.**
(XLSX)

**S3 Data. Numerical data used in all figures.**
(XLSX)

**S1 Text. Supporting Methods.**
(DOCX)

**S1 Table.** (A) Primers used for PCR amplification to target the bacterial 16S rRNA gene, quantification (qPCR). * indicates the reference numbers. (B) The relative abundances of 15 representative bacterial strains isolated from the guts conventional flies 5 dpe to 28˚C and 10˚C. *The strains isolated from the gut of *B. dorsalis* in this study are designated with numbers, e.g., BD473 according to our previous publication, with only one new strain *Serratia marcescens* isolated in this study represented as BD164.
(DOCX)

## Acknowledgments

We thank the team at American journal experts for critically revising the English language of this manuscript.

## Author Contributions

**Conceptualization:** Muhammad Fahim Raza, Hongyu Zhang.

**Data curation:** Muhammad Fahim Raza, Zhaohui Cai, Shuai Bai.

**Formal analysis:** Muhammad Fahim Raza, Umar Anwar Awan.

**Funding acquisition:** Hongyu Zhang.

**Investigation:** Muhammad Fahim Raza, Zhichao Yao, Weiwei Zheng.

**Methodology:** Muhammad Fahim Raza, Yichen Wang, Zhenyu Zhang.

**Resources:** Hongyu Zhang.

**Supervision:** Hongyu Zhang.

**Validation:** Zhaohui Cai, Hongyu Zhang.

**Writing – original draft:** Muhammad Fahim Raza, Yichen Wang, Hongyu Zhang.

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
