## [Decision Letter · Decision Letter 0]

25 Oct 2019

Dear Mr Raza,

Thank you very much for submitting your manuscript "Gut Microbiota Promotes Host Resistance to Low-Temperature Stress by Stimulating its Arginine and Proline Metabolism Pathway in Adult Bactrocera dorsalis" (PPATHOGENS-D-19-01692) for review by PLOS Pathogens. Your manuscript was fully evaluated at the editorial level and by independent peer reviewers. The reviewers appreciated the attention to an important problem, but raised some substantial concerns about the manuscript as it currently stands. These issues must be addressed before we would be willing to consider a revised version of your study. We cannot, of course, promise publication at that time.

We therefore ask you to modify the manuscript according to the review recommendations before we can consider your manuscript for acceptance. Your revisions should address the specific points made by each reviewer.

(1) A letter containing a detailed list of your responses to the review comments and a description of the changes you have made in the manuscript. Please note while forming your response, if your article is accepted, you may have the opportunity to make the peer review history publicly available. The record will include editor decision letters (with reviews) and your responses to reviewer comments. If eligible, we will contact you to opt in or out.

(2) Two versions of the manuscript: one with either highlights or tracked changes denoting where the text has been changed; the other a clean version (uploaded as the manuscript file).

Additionally, to enhance the reproducibility of your results, PLOS recommends that you deposit your laboratory protocols in protocols.io, where a protocol can be assigned its own identifier (DOI) such that it can be cited independently in the future. For instructions see http://journals.plos.org/plospathogens/s/submission-guidelines#loc-materials-and-methods

We hope to receive your revised manuscript within 60 days. If you anticipate any delay in its return, we ask that you let us know the expected resubmission date by replying to this email. Revised manuscripts received beyond 60 days may require evaluation and peer review similar to that applied to newly submitted manuscripts.

[LINK]

Sincerely,

Guy Tran Van Nhieu

Section Editor

PLOS Pathogens

Guy Tran Van Nhieu

Section Editor

PLOS Pathogens

Kasturi Haldar

Editor-in-Chief

PLOS Pathogens

orcid.org/0000-0001-5065-158X

Grant McFadden

Editor-in-Chief

PLOS Pathogens

orcid.org/0000-0002-2556-3526

Reviewer's Responses to Questions

**Part I - Summary**

Reviewer #1: The relationship between the microbiome and it multicellular host, whether vertebrate or invertebrate, is a very hot topic in biology. Novel interactions are coming to light, illustrating that hosts and resident bacteria work closely together in a variety of hitherto unknown ways.

In this highly original, rigorous and thorough study, Raza et al., show that a member of the bacterial community of the oriental fruit fly confers the ability to resist cold stress to its host, and decipher the metabolic pathway that is enhanced by the symbiont.

It was a pleasure to read this thoughtful and comprehensive study. After establishing that K. michiganensis enables flies to survive cold stress, authors embarked on a quest to find the mechanism whereby this ability is enabled. They performed energetic analyses, metabolomics and transcriptomics, to identify that the arginine and proline metabolic pathways are enhanced in symbiotic flies. Importantly, they performed a series of manipulative experiments where axenic flies were inoculated with K. michiganensis. These experiments show clearly the effect of the symbiont on the metabolic pathways, and its long term effect on survival.

The findings are highly novel and rigorously analyzed, and this is an exciting study.

Reviewer #2: This paper investigates whether the gut microbiota is involved in B. dorsalis survival during cold stress. Out of multiple microbial strains tested, K. michiganensis extended fly survival in cold to a level similar to conventionally-colonized flies. Transcriptomic and metabolomic studies suggested that K. michiganensis influences the host's arginine and proline pathways. Subsequent studies mimicking an axenic state through RNAi against arginine and proline genes reduced the survival of conventional flies under cold stress, and injecting arginine and proline into axenic flies improved median survival under cold stress. This is an interesting study that demonstrates the importance of arginine and proline metabolism in B. dorsalis cold stress survival. I have only a few criticisms.

Reviewer #3: The work "Gut microbiota promotes host resistance to low-temperature stress by stimulating its arginine and proline metabolism pathway in adult Bactrocera dorsalis" investigates the role of specific symbiotic bacteria on host physiology and their contribution to survival to extreme environmental conditions. The oriental fruit fly Bactrocera dorsalis is highly resistant to long-term low-temperature stress that facilitate its survival in novel environments. This is an interesting trait of this insect's biology since the oriental fruit fly is a pest that causes substantial lost of cultivated crops in Asia and damage 250 host fruits and vegetables worldwide. The authors identified the fly's symbiotic bacterium K. michiganensis BD177 as responsible for the survival of the adult B. dorsalis at low temperature which allows to the pest to spread in different environments. The authors performed metabolomic and transcriptomic analysis on conventional, axenic and K. michiganensis recolonized axenic flies to determine the bacterium factors that promote the survival. The authors concluded that K. michiganensis is necessary to promote arginine and proline metabolisms that in turn are important metabolites to promote longevity at low temperature. While this work has high biological and biothechonolgical relevance since it highlights one of the possible biological factors that could be targeted to reduce the competitiveness of this pest and potentially others, on the other hand it identified a mechanism of response to cold stress that was already described in other insects ("Physiological basis for low-temperature survival and storage of quiescent larvae of the fruit fly Drosophila melanogaster"Koštá et al. Sci. Rep 2016) . The authors did not try to analyze the mechanisms by which the proline or arginine serum accumulation might promote a better survival at low temperature neither how the host bacterium control the host metabolism. As it stands this work has the merit to have identified that the metabolic changes linked to temperature adaptation in the Tephritidae B. dorsalis are caused by the symbiotic bacterium K. michiganensis but it does nt attempt any mechanistic studies which i believe is expected by the readership of Plos Pathogens. The metabolic factors identified had been already linked to temperature resistance in insects. Moreover, the contributions of the microbiota to host physiology also have already been reported in different species (for a review Capo et al, 2019; or specific papers Behar et al, 2008; Shin et al, 2011).

So while the paper is interesting and well controlled relatively to the claims made, I think that is a bit too descriptive and although mechanisms are hypothesized, no experiment is made to try to prove them. Moreover in the analysis of the species that might contribute to longevity at low temperatures, some species should have been tested in combination with K. michiganensis but were instead ignored, underestimating the impact of microbes interaction on the host physiology. Therefore more experiments into mechanisms and some clarifications should be provided prior of publication (see major and minor revisions below)

**Part II – Major Issues: Key Experiments Required for Acceptance**

Reviewer #1: na

Reviewer #2: Past studies with D. melanogaster and D. suzukii have shown that microbivorism can enhance host health and development. These effects can be achieved by live microbes, or by dead microbes that are given to the host continually and at levels that effectively mimic the propagation and continual presence of live microbial growth. In Bing et al. (2018), live microbes enhance D. suzukii development, and heat-killed microbes also rescue D. suzukii development in a dose-dependent manner. Similarly, in Yamada et al. (2015) and Keebaugh et al. (2018), the lifespan-extending effect of microbes can be fully mimicked by dead microbes that are continually given to flies at a level that is equivalent to what an initial inoculum of live microbial growth would have produced. Essentially, microbes that grow better are better at rescuing host health under nutritional stress. This nutritional aspect of microbes should therefore be tested as a potential factor in host–microbe studies, especially when using hosts that naturally feed on food substrates that contain microbial growth, like fruit and vinegar flies.

Data within this paper support the idea that K. michiganensis grows well in the cold because it reaches CFU levels equivalent to the total microbial load of conventional B. dorsalis. This suggests that microbial growth alone could be the main factor in how microbes influence cold survival. One way to test this hypothesis might be to provide each live microbe at equivalent initial doses, and then monitor the growth of each microbe over time to determine if host benefits are specific to K. michiganensis, or if general microbial growth can stimulate host metabolism in a way that is beneficial during cold stress. Further, the authors should properly test if heat-killed K. michiganensis can mimic live K. michiganensis in survival trials. Heat-killed K. michiganensis should be continually given to the flies at levels equivalent to live microbial growth in order to determine whether live microbes actively influence the host in a way that boosts cold survival. Directly quantifying bacterial growth rates at different temperatures might also be informative.

Although the manuscript focuses on the "gut microbiota," the data doesn't convincingly rule out contributions from other sources. I assume that the axenic treatment doesn't simply remove the gut bacteria, but also eliminates it from all niches (on the fly, in the fly, and in the fly environment/food). How does one show that the causal factor is the gut microbiome in particular? Do these flies have a well-established, stable gut microbiome?

Again, overall I am positive about this work but the central message of the report should be revised accordingly depending on how the above factors are addressed.

Reviewer #3: Major revisions:

1)Arginine and proline pathways are dependent on the TCA cycle and a previous study has linked mitochondrial activity to the arginine and proline induction in Drosophila's resistance to low temperature ("Physiological basis for low-temperature survival and storage of quiescent larvae of the fruit fly Drosophila melanogaster"Koštá et al. Sci. Rep 2016), I believe that mitochondria status should be assessed in this fly by transmitted electron microscopy or mitotracker staining followed by fluorescent microscopy and measurement of mitochondria morphology to determine whether the metabolic changes could be linked to the mitochondria activity.Thus mitochondria status should be tested on conventional, axenic and K. michiganensis recolonized flies to assess whether the bacterium affect mitochondria activity in general. Moreover, to measure the mitochondria functionality and also to better define the metabolic mechanisms that promote the host survival , ATP measurements should be taken in all conditions.

2) The authors observed that high TG in the hemolymph is a metabolic features of conventional resistant flies to the temperature stress. The level of TG or TG storage should be assessed in the fly fat body to understand whether fat synthesis or storage consumptions are also responsible for the survival mechanisms.

3)Are lipid synthetic genes or catabolic genes up-regulated in the transcriptomic analysis to justify this TG increase? My apologies if this was not clear to me.

4) Injection of flies with linoleic acid and arachidonic acid does not necessarily recapitulate the metabolic phenotype linked to temperature stress resistance. TG are made of different saturated and unsaturated fatty acid chains. What kind of TG are in the hemolymph? Because the exact types should be injected to rule out whether TG in the hemolymph are responsible or not for the survival. A diet on holidic media supplied with different fatty acid chains should be given to the flies in different conditions before coming to the conclusion that TG have no impact on the low-temperature resilience of the host.

5) Is this bacterium important to provide resistance to other types of stress? Other stressing should be tested to verify whether it specifically helps the host to activate temperature stress response pathways or also control responses to other common stresses such as UV exposure.

6) Does injection with arginine and proline together improve the viability?

7) And does ds RNA-mediated silencing of target genes that affect both pathways reduce the survival?

**Part III – Minor Issues: Editorial and Data Presentation Modifications**

Reviewer #1: The ms., is very well written and despite the complexity, easy to follow. There are a few very minor points that need attention:

Line 73- “The mechanisms underlying cold acclimation responses are the main focus of much research in insect species”. References are needed to back this up. E.g.:

Overgaard, J., MacMillan, H.A., 2017. The Integrative Physiology of Insect Chill Tolerance, Annual Review of Physiology, Vol 79, pp. 187-208.

Teets, N.M., Denlinger, D.L., 2013. Physiological mechanisms of seasonal and rapid cold-hardening in insects. Physiological Entomology 38, 105-116.

110- [31]

112- figure 2- but no reference to figure 1? Use fig 1 in the methods section and renumber the figures accordingly.

121- figure 3- extend the x axis to show the full extent of survival (as in the other survival figures), not just the first 6 days, or explain why not.

Reviewer #2: Other suggestions:

The fly diet should be described in better detail.

The heat-killed dosing protocol should be described in better detail.

The authors show that oral antibiotics do not influence host survival under normal rearing conditions, but what about treatment conditions (in the cold)?

The reinfection protocol is not normalized by bacterial load (CFUs) because a general OD measure was used. Thus, B. dorsalis were likely given very different levels of microbes at the start of monoculture experiments. If microbial growth alone is influential on cold survival, then different inoculum sizes could influence microbial growth dynamics and ultimately host survival. These experiments should be done in a controlled manner in which the initial CFU is normalized across microbial strains.

There were CFUs detected in flies that were supposed to be axenic or given heat-killed microbes. Was this contamination? Unless there is an explanation for this, the presence of live microbes is a confounding factor in all experiments using flies that should be absent of live microbial growth.

Reviewer #3: Minor issues:

1) Figure made of one or two panels could be combined in single figure to facilitate the reader.

2) The experimental details for the metabolomic analysis should be provided (e.g. how many ul of hemolymph, where they normalized to number of flies? weight? protein of the carcasse? )as well as for the transcriptomic (amount of RNA, kit used, seq platform etc.)

PLOS authors have the option to publish the peer review history of their article (what does this mean?). If published, this will include your full peer review and any attached files.

Reviewer #1: No

Reviewer #2: No

Reviewer #3: No

---

## [Decision Letter · Decision Letter 1]

17 Feb 2020

Dear Prof. Zhang,

Thank you very much for submitting your manuscript "Gut Microbiota Promotes Host Resistance to Low-Temperature Stress by Stimulating its Arginine and Proline Metabolism Pathway in Adult Bactrocera dorsalis" for consideration at PLOS Pathogens. As with all papers reviewed by the journal, your manuscript was reviewed by members of the editorial board and by several independent reviewers. The reviewers appreciated the attention to an important topic. Based on the reviews, we are likely to accept this manuscript for publication, providing that you modify the manuscript according to the review recommendations. Specifically, reviewer 2 request that some points are discussed / clarified.

Sincerely,

Guy Tran Van Nhieu

Section Editor

PLOS Pathogens

Guy Tran Van Nhieu

Section Editor

PLOS Pathogens

Kasturi Haldar

Editor-in-Chief

PLOS Pathogens

orcid.org/0000-0001-5065-158X

Michael Malim

Editor-in-Chief

PLOS Pathogens

orcid.org/0000-0002-7699-2064

Reviewer Comments (if any, and for reference):

Reviewer's Responses to Questions

**Part I - Summary**

Reviewer #1: Authors have responded to queries in a very thorough and satisfactory manner. the additional data strengthen the ms. considerably, and i believe it is an important contribution. Ms. should be accepted.

on a personal note, i would like to convey my concern to the authors re the CoronaVirus epidemic, and wish them and their families safety and good health.

Boaz Yuval

Reviewer #2: This revised manuscript examines whether the gut microbiota is involved in B. dorsalis survival during cold stress. K. michiganensis extended fly survival in cold to a level similar to conventionally-colonized flies. Transcriptomic and metabolomic studies suggested that K. michiganensis influences the host's arginine and proline pathways. Subsequent studies mimicking an antibiotic-treated state through RNAi against arginine and proline metabolism genes reduced the survival of conventional flies under cold stress, and injecting arginine and proline into axenic flies improved median survival under cold stress. The revised study adds substantial data to bolster the relationship between the gut microbiota, host metabolism, and mitochondrial function to resist cold stress. Although the additional studies raise new questions that could be cleaned up a little further experimentally, I don't see them as being necessary new experiments. Thus, I have raised them only as minor concerns.

Reviewer #3: The manuscript "Gut Microbiota Promotes Host Resistance to Low-Temperature Stress by Stimulating its

2 Arginine and Proline Metabolism Pathway in Adult Bactrocera dorsalis"authored by Raza et al, dissect the biochemical and genetic contributions of the microbiota to the host fitness in the oriental fruit fly Bactrocera dorsalis. The relevance of the study is to have established that the microbiota provides host resistance to low-temperature stress in B. dorsalis by stimulating its arginine and proline

metabolism pathway. The discovery is important because B. dorsals is an invasive pest of fruit crops

that exhibit a unique form of resistance to mild or even extreme temperature

conditions. This resistance may facilitate the survival of this pest in novel environments and increase the pest; negative impact on the crops production. The author identified the mechanisms by which the resistance to cold occurs. he work is original relatively to this specific insect of agricultural importance, however the mechanisms by which the cold resistance can be regulated by the microbiota in insects was already described in previous study h ("Physiological basis for low-temperature survival and storage of quiescent larvae of the fruit fly Drosophila melanogaster "Koštá et al. Sci. Rep 2016).

This version of the manuscript has greatly improved compared to the first one and more controls and further mechanistic details have been added following the reviewers comment. The authors have satisfied this reviewer concerns and therefore in my opinion the manuscript can be accepted for publication.

**Part II – Major Issues: Key Experiments Required for Acceptance**

Reviewer #1: na

Reviewer #2: n/a

Reviewer #3: The authors have addressed my previous major issues.

**Part III – Minor Issues: Editorial and Data Presentation Modifications**

Reviewer #1: na

Reviewer #2: The authors have added additional data that bolster the conclusions. The new data are substantial, and while other studies could be added to further support the findings (i.e. additional controls), I don't think they're absolutely necessary. The authors should still consider conducting these studies or acknowledging them by modification of the text.

1) Antibiotic treatment has been shown to affect mitochondrial function in various animals. Hence, although the authors' new data on mitochondria morphology is consistent with their overall interpretations, it might be valuable to repeat key studies using a truly axenic protocol (without using antibiotics). I'm not sure that's possible with their system. Nonetheless, reinfection with K. michiganensis rescues the effect, so I don't think this is a huge concern.

2) The authors could also consider looking at mitochondria after amino acid injection (and/or dsRNA manipulations). This would greatly strengthen the findings.

3) From the types of studies that were done, it's hard to say whether injection of proline and arginine together results in an additive or synergistic effect. This doesn't have to be determined here, but could be discussed. I may have missed it, but did the authors also test other amino acids, as another control on the specificity for these amino acid pathways? I think that experiment would be a really nice control to have, but I don't think it's absolutely needed.

4) The use of heat-killed bacteria greatly helps support the authors' conclusions. The maximum provision of heat-killed bacteria was 10^9 at each food change. This is compared to the single inoculation of live bacteria of 10^8. My question here is, with single live inoculations, how much environmental bacterial growth occurs and is the fly constantly receiving reinoculations from live bacteria from the environment? If so, are the bacterial CFUs that the hosts are exposed to equivalent to the 10^9 at each food change? From my calculations from other publications, it seems that bacterial growth rates can sometimes require repeated exposures of 100-fold more dead bacteria to match the rapid growth rates of live bacteria. The authors should be quite clear that they are using roughly equivalent amounts to truly justify their interpretations.

Reviewer #3: I still believe that the Figures could be filled more since most of them (e.g. Figure 1 and 2) still comprise only 2 panels and it is awkward.

PLOS authors have the option to publish the peer review history of their article (what does this mean?). If published, this will include your full peer review and any attached files.

Reviewer #1: No

Reviewer #2: No

Reviewer #3: No
---

## [Editor Report · Decision Letter 2]

28 Feb 2020

Dear Prof. Zhang,

We are pleased to inform you that your manuscript 'Gut Microbiota Promotes Host Resistance to Low-Temperature Stress by Stimulating its Arginine and Proline Metabolism Pathway in Adult Bactrocera dorsalis' has been provisionally accepted for publication in PLOS Pathogens.

Best regards,

Guy Tran Van Nhieu

Section Editor

PLOS Pathogens

Guy Tran Van Nhieu

Section Editor

PLOS Pathogens

Kasturi Haldar

Editor-in-Chief

PLOS Pathogens

orcid.org/0000-0001-5065-158X

Michael Malim

Editor-in-Chief

PLOS Pathogens

orcid.org/0000-0002-7699-2064
---

## [Editor Report · Acceptance letter]

3 Apr 2020

Dear Prof. Zhang,

We are delighted to inform you that your manuscript, "Gut Microbiota Promotes Host Resistance to Low-Temperature Stress by Stimulating its Arginine and Proline Metabolism Pathway in Adult *Bactrocera dorsalis*," has been formally accepted for publication in PLOS Pathogens.

Best regards,

Kasturi Haldar

Editor-in-Chief

PLOS Pathogens

orcid.org/0000-0001-5065-158X

Michael Malim

Editor-in-Chief

PLOS Pathogens

orcid.org/0000-0002-7699-2064